# An Improved Algorithm for
# Adversarial Linear Contextual Bandits via Reduction

**Tim van Erven**[*]  **Jack Mayo**[†]  **Julia Olkhovskaya**[‡]  **Chen-Yu Wei**[§]

## Abstract

We present an efficient algorithm for linear contextual bandits with adversarial losses and stochastic action sets. Our approach reduces this setting to misspecification-robust adversarial linear bandits with fixed action sets. Without knowledge of the context distribution or access to a context simulator, the algorithm achieves $\widetilde{\mathcal{O}}(\min\{d^2\sqrt{T}, \sqrt{d^3 T \log K}\})$ regret and runs in $\text{poly}(d, C, T)$ time, where $d$ is the feature dimension, $C$ is an upper bound on the number of linear constraints defining the action set in each round, $K$ is an upper bound on the number of actions in each round, and $T$ is number of rounds. This resolves the open question by Liu et al. (2023) on whether one can obtain $\text{poly}(d)\sqrt{T}$ regret in polynomial time independent of the number of actions. For the important class of combinatorial bandits with adversarial losses and stochastic action sets where the action sets can be described by a polynomial number of linear constraints, our algorithm is the first to achieve $\text{poly}(d)\sqrt{T}$ regret in polynomial time, while no prior algorithm achieves even $o(T)$ regret in polynomial time to our knowledge. When a simulator is available, the regret bound can be improved to $\widetilde{\mathcal{O}}(d\sqrt{L^\star})$, where $L^\star$ is the cumulative loss of the best policy.

## 1 Introduction

We consider the following linear contextual bandit problem: At each round $t = 1, \ldots, T$, the environment generates a hidden loss vector $\theta_t \in \mathbb{R}^d$ and an action set $\mathcal{A}_t \subset \mathbb{R}^d$. The learner observes $\mathcal{A}_t$, selects an action $a_t \in \mathcal{A}_t$, and incurs loss $a_t^\top \theta_t$. The goal is to compete with the best fixed policy—defined as a mapping from an action set to an element in it. This setting generalizes the classical linear bandit model by allowing the action sets $\mathcal{A}_t$ to vary stochastically over time. Crucially, each $\mathcal{A}_t$ encodes the *context* based on which the learner makes decisions. In this work, $\mathcal{A}_t$ is called *context* or *action set* interchangeably.

This framework is applicable in settings such as healthcare and recommendation systems, where decisions must be made conditional on context. Prior work on contextual bandits has studied a variety of assumptions on how losses and contexts are generated. While much of the literature assumes i.i.d. losses and arbitrarily chosen action sets (for which a well-known algorithm is LinUCB (Li et al., 2010)), we focus on the complementary regime: the action sets are drawn i.i.d. from a fixed distribution $\mathcal{D}$, while the losses may be chosen adversarially.

A first computationally efficient algorithm for this setting was proposed by Neu and Olkhovskaya (2020) under the assumption that the context (i.e., action set) distribution is *known*. Since an action set is a subset of $\mathbb{R}^d$ (i.e., it lies in the space $2^{\mathbb{R}^d}$), the distribution over action sets is in the space

---

[*]University of Amsterdam. Email: `tim@timvanerven.nl`

[†]University of Amsterdam; Kurtos.ai. Email: `jackjamesmayo@gmail.com`

[‡]Delft University of Technology. Email: `julia.olkhovskaya@gmail.com`

[§]University of Virginia. Email: `chenyu.wei@virginia.edu`

39th Conference on Neural Information Processing Systems (NeurIPS 2025).

Table 1: Comparison with state-of-the-art results in adversarial linear contextual bandits. $d$ is the feature dimension, $K$ is an upper bound on the number of actions, and $C$ is an upper bound on the number of linear constraints to describe the convex hull of each action set. It holds that $C \leq K + 1$ and in many combinatorial problems we have $C = \text{poly}(d)$ and $K = 2^{\Omega(d)}$. The run time of the linear optimization oracle of Neu and Valko (2014) is bounded by $\text{poly}(d, C)$ but could also be smaller.

| Algorithm | Regret (omitting $\log(dT)$ factors) | Computation | Simulator | Feedback |
|---|---|---|---|---|
| Schneider and Zimmert (2023) | $\sqrt{dT}$ for the special case $\mathcal{A}_t \subseteq \{\mathbf{e}_1, \ldots, \mathbf{e}_d\}$ | $\text{poly}(d, T)$ | no | bandit |
| Neu and Valko (2014) | $(dT)^{2/3}$ | $\text{poly}(d, T)$ plus $T$ oracle calls | no | semi-bandit |
| Dai et al. (2023) | $\min\{d\sqrt{T}, \sqrt{dT \log K}\}$ | $\text{poly}(d, K, T)$ | yes | bandit |
| Liu et al. (2023) | $d\sqrt{T}$ | $K \cdot T^{\Omega(d)}$ | no | bandit |
| Liu et al. (2023) | $d^2\sqrt{T}$ | $\text{poly}(d, K, T)$ | no | bandit |
| Ours | $\min\{d^2\sqrt{T}, \sqrt{d^3 T \log K}\}$ | $\text{poly}(d, C, T)$ | no | bandit |
| Ours | $d\sqrt{L^\star}$ | $\text{poly}(d, C, T)$ | yes | bandit |

$\Delta(2^{\mathbb{R}^d})$, which is generally intractable to represent efficiently. This assumption was later removed by subsequent efficient algorithms (Luo et al., 2021; Sherman et al., 2023; Dai et al., 2023; Liu et al., 2023). In the setting where the learner has access to a simulator that can generate free contexts (i.e., the learner is able to sample contexts from $\mathcal{D}$ as many times as they want without incurring cost), Dai et al. (2023) shows that a near-optimal regret bound of $\widetilde{\mathcal{O}}(\min\{d\sqrt{T}, \sqrt{dT \log K}\})$ is achievable, where $K = \max_t |\mathcal{A}_t|$. When the learner has neither knowledge of the context distribution nor simulator access to random context samples, the best known results are by Liu et al. (2023): they provide an algorithm with near-optimal regret $\widetilde{\mathcal{O}}(d\sqrt{T})$ with run time $T^{\Omega(d)}$ and another algorithm with regret $\widetilde{\mathcal{O}}(d^2\sqrt{T})$ with run time $\text{poly}(d, K, T)$. Notably, while the regret bound of this last algorithm is independent of the number of actions $K$, its computational complexity scales polynomially in $K$. In fact, this is the case for all previous algorithms as well (Luo et al., 2021; Sherman et al., 2023; Dai et al., 2023; Liu et al., 2023, 2024b). This makes them unsuitable for many important combinatorial problems (e.g., $m$-set, shortest path, flow, bipartite matching), where $K$ is usually exponentially large in the dimension $d$.

Our work gives the first algorithm whose computational complexity does not explicitly scale with the number of actions, making adversarial linear contextual bandits applicable to a much wider range of problems. Without simulator access, our method achieves regret $\widetilde{\mathcal{O}}(\min\{d^2\sqrt{T}, \sqrt{d^3 T \log K}\})$ and with simulator access this can be improved to $\widetilde{\mathcal{O}}(d\sqrt{L^\star})$, where $L^\star = O(T)$ is the cumulative loss of the best policy. Our algorithm runs in $O(d, C, T)$ time, where $C$ is the number of linear constraints to describe the convex hull of each action set. Notice that $C \leq K + 1$ in general, as the convex hull of an action set of size $K$ can be written as a linear program with at most $K + 1$ constraints. On the other hand, in many combinatorial problems, $C = \text{poly}(d)$ while $K = 2^{\Omega(d)}$. For example, in a shortest path problem with $d$ edges, the set of all paths can be described by a linear program with $O(d)$ constraints, while the number of paths could be of order $2^{\Omega(d)}$. For combinatorial problems with stochastic action sets and adversarial losses, we are only aware of Neu and Valko (2014) who studied the case where the learner has *semi-bandit* feedback. Their algorithm achieves $\widetilde{\mathcal{O}}((dT)^{2/3})$ regret with one call to the linear optimization oracle of the action set per round. Compared to their work, our work weakens the assumption on the feedback (from semi-bandits to full-bandits) and improves the regret bound, but our method is computationally heavier—there are action sets where linear optimization can be solved in polynomial time while having an exponentially large number of constraints, such as spanning trees. How to further improve our computational complexity to match theirs is left as future work. It also remains open how to efficiently achieve the near-optimal bound $\widetilde{\mathcal{O}}(\min\{d\sqrt{T}, \sqrt{dT \log K}\})$ without simulators, even when the learner is willing to pay $\text{poly}(d, K, T)$ computation. This has only been resolved in the special case where action sets

are subsets of the standard basis $\{\mathbf{e}_1, \ldots, \mathbf{e}_d\}$, which is also known as the sleeping bandits problem (Schneider and Zimmert, 2023).

Our result is achieved by establishing a novel and computationally efficient reduction from *adversarial linear contextual bandits* to *adversarial linear bandits with misspecification*. Linear bandits can be viewed as linear contextual bandits with a fixed context and is therefore less challenging than linear contextual bandits. This reduction scheme is related to the work of Hanna et al. (2023), which reduces stochastic linear contextual bandits (with both stochastic contexts and stochastic losses) to misspecified stochastic linear bandits. Compared to Hanna et al. (2023), we make two key advances: First, our reduction is polynomial-time, and we also provide a polynomial-time algorithm for the problem we reduce to, whereas Hanna et al. (2023) focus solely on the regret bound and it remains open how their reduction can be implemented in polynomial time. Second, we study the strictly more general setting with adversarial losses. A more detailed comparison with Hanna et al. (2023) is provided in Appendix A.

Finally, regarding results in terms of $L^\star$, we remark that our bound of $\widetilde{\mathcal{O}}(d\sqrt{L^\star})$ can be specialized to the setting of Olkhovskaya et al. (2023) (a slightly different formulation of adversarial contextual bandits). We then obtain a slightly worse rate but, importantly, remove their unrealistic assumption that the context distribution is log-concave.

## 2 Problem Description and Main Results

**Notation** Suppose $\mathcal{A} \subset \mathbb{R}^d$ is a set of vectors. Then the convex hull of $\mathcal{A}$ is denoted as $\mathrm{conv}(\mathcal{A}) = \{x = \sum_{i=1}^k \lambda_i a_i : k \in \mathbb{N}, \sum_{i=1}^k \lambda_i = 1, \lambda_i \geq 0, a_i \in \mathcal{A}\}$. If $\mathcal{A}$ is a polytope, we denote its set of vertices as $\mathrm{vert}(\mathcal{A})$, and denote the normal cone of $\mathcal{A}$ at a vertex $v \in \mathrm{vert}(\mathcal{A})$ as $\mathcal{N}(\mathcal{A}, v) = \{y \in \mathbb{R}^d : \max_{x \in \mathcal{A}} \langle y, x - v \rangle \leq 0\}$.

### 2.1 Linear Contextual Bandits

For simplicity, we consider an oblivious adversary.[5] Before any interaction with the learner, the adversary secretly chooses $T$ loss vectors $(\theta_t)_{t \in [T]}$ where $\theta_t \in \mathbb{R}^d$ for all $t$. For each round $t = 1, 2, \ldots, T$, an action set $\mathcal{A}_t \subset \mathbb{R}^d$ is drawn according to $\mathcal{A}_t \overset{\mathrm{i.i.d.}}{\sim} \mathcal{D}$ and revealed to the learner. The learner then chooses an action $a_t \in \mathcal{A}_t$ and observes loss $\ell_t \in [0, 1]$ with $\mathbb{E}_t[\ell_t] = a_t^\top \theta_t$, where $\mathbb{E}_t$ denotes the expectation conditioned on the history up to time $t - 1$ and $a_t$. The expected regret against a fixed policy $\pi \in \Pi$ is defined as

$$\mathrm{Reg}_T(\pi) = \mathbb{E}\left[\sum_{t=1}^T \langle a_t - \pi(\mathcal{A}_t), \theta_t \rangle\right],$$

where policy $\pi$ maps an action set $\mathcal{A} \subset \mathbb{R}^d$ to a point $\pi(\mathcal{A}) \in \mathrm{conv}(\mathcal{A})$.[6] Note that the policy that minimizes the total expected loss is

$$\arg\min_{\pi \in \Pi} \mathbb{E}\left[\sum_{t=1}^T \langle \pi(\mathcal{A}_t), \theta_t \rangle\right] = \arg\min_{\pi \in \Pi} \mathbb{E}_{\mathcal{A} \sim \mathcal{D}}\left[\left\langle \pi(\mathcal{A}), \sum_{t=1}^T \theta_t \right\rangle\right],$$

which is attained by the policy $\pi(\mathcal{A}) = \arg\min_{a \in \mathcal{A}} \langle a, \sum_{t=1}^T \theta_t \rangle$. Thus, to minimize the expected regret, it suffices to compare the learner to the class of linear classifier policies

$$\Pi_{\mathrm{lin}} = \left\{\pi_\phi \mid \phi \in \mathbb{R}^d\right\} \text{ where } \pi_\phi(\mathcal{A}) = \arg\min_{a \in \mathcal{A}} \langle a, \phi \rangle, \tag{1}$$

where ties can be broken arbitrarily.

---

[5]There are two kinds of results we can get: 1) regret compared to the best policy in the full policy set $\Pi$ under oblivious adversary, and 2) regret compared to the best policy in the linear policy set $\Pi_{\mathrm{lin}}$ under adaptive adversary. To achieve 2), it suffices to use misspecification-robust linear bandit algorithms with high-probability bounds in our reduction. This can be achieved by standard techniques of getting high-probability bounds (Lee et al., 2020; Zimmert and Lattimore, 2022). To simplify the exposition, we only focus on the first case.

[6]Defining $\pi(\mathcal{A}) \in \mathrm{conv}(\mathcal{A})$ instead of $\pi(\mathcal{A}) \in \mathcal{A}$ only makes the guarantee more general and simplifies the notation. Equivalently, it defines a randomized policy: to execute $\pi(\mathcal{A})$, sample $a \in \mathcal{A}$ such that $\mathbb{E}[a] = \pi(\mathcal{A})$.

## 2.2 Linear Bandits and $\epsilon$-Misspecified Linear Bandits

The adversarial linear bandit problem with oblivious adversary is the case in which the adversary decides $(\theta_t)_{t\in[T]}$ before any interaction with the learner. The learner is given a fixed action set $\widehat{\Omega} \subset \mathbb{R}^d$. At every round $t \in [T]$, the learner chooses an action $y_t \in \widehat{\Omega}$ and receives $c_t \in [0,1]$ as feedback with $\mathbb{E}_t[c_t] = \langle y_t, \theta_t \rangle$. The regret with respective to a fixed action $y \in \widehat{\Omega}$ is defined as

$$\text{Reg}_T(y) = \mathbb{E}\left[\sum_{t=1}^{T} \langle y_t - y, \theta_t \rangle\right].$$

A *misspecified* linear bandit problem is the case where the learner, instead of receiving an unbiased sample of $\langle y_t, \theta_t \rangle$ as feedback, receives $c_t$ with $|\mathbb{E}_t[c_t] - \langle y_t, \theta_t \rangle| \le \epsilon$ for some $\epsilon$ known to the algorithm.

## 2.3 Results Overview

In this section, we present a general framework that can reduce the adversarial contextual bandit problem to a misspecification-robust linear bandit algorithm defined as the following.

**Definition 1** ($\alpha$-misspecification-robust adversarial linear bandit algorithm)**.** *A $\alpha$-misspecification-robust linear bandit algorithm over action set $\widehat{\Omega} \subset \mathbb{R}^d$ has the following property: with a given $\epsilon > 0$ and the guarantee that every time the learner chooses $y_t \in \widehat{\Omega}$, the loss received $c_t \in [0,1]$ satisfies*

$$|\mathbb{E}_t[c_t] - \langle y_t, \theta_t \rangle| \le \epsilon,$$

*the algorithm ensures*

$$\mathbb{E}\left[\sum_{t=1}^{T} \langle y_t, \theta_t \rangle\right] \le \min_{y\in\widehat{\Omega}} \sum_{t=1}^{T} \langle y, \theta_t \rangle + \widetilde{O}\left(d\sqrt{T} + \alpha\sqrt{d}\epsilon T\right). \tag{2}$$

Notice that there is an $\alpha$ parameter in Definition 1 that specifies the dependence of the regret on the misspecification level $\epsilon$. It is known that $\alpha = 1$ is the statistically optimal dependence. However, for specific algorithms, we might have $\alpha > 1$.

We establish the following reduction:

**Theorem 1.** *Given access to an $\alpha$-misspecification-robust adversarial linear bandit algorithm, we can achieve $\min_{\pi\in\Pi} \text{Reg}_T(\pi) \le \widetilde{O}(d\sqrt{T} + \alpha d\sqrt{T\log K})$ in adversarial linear contextual bandits without access to a simulator.*

We remark that the $\alpha d\sqrt{T\log K}$ term in Theorem 1 comes from the misspecification term $\alpha\sqrt{d}\epsilon T$ in (2). When the learner has access to a simulator that generates free contexts, the $\epsilon$ can be made arbitrarily small, allowing us to achieve the optimal $d$ dependence. This will be discussed in Section 5.

## 3 Reduction from Linear Contextual Bandits to Linear Bandits

Let $\pi$ denote a policy, which maps any given action set $\mathcal{A}$ to a randomized action in $\text{conv}(\mathcal{A})$. Let $\Pi$ denote the set of all possible policies. We define the following map

$$\Psi(\pi) = \mathbb{E}_{\mathcal{A}\sim\mathcal{D}}[\pi(\mathcal{A})],$$

which is the mean action of $\pi$. Applying $\Psi$ to all $\pi \in \Pi$, the set $\Omega = \{\Psi(\pi) \mid \pi \in \Pi\}$ is induced. Note that, under this map, the expected loss under actions drawn such that $\mathbb{E}[a_t] = \pi(\mathcal{A}_t)$ may be written as

$$\mathbb{E}[\langle a_t, \theta_t\rangle] = \mathbb{E}_{\mathcal{A}_t\sim\mathcal{D}}[\langle \pi(\mathcal{A}_t), \theta_t\rangle] = \langle \Psi(\pi), \theta_t \rangle.$$

Accordingly, if the learner draws $a_t$ from policy $\pi_t$ in round $t$, the expected regret may be written as

$$\text{Reg}_T(\pi) = \mathbb{E}\left[\sum_{t=1}^{T} \langle a_t - \pi(\mathcal{A}_t), \theta_t\rangle\right] = \mathbb{E}\left[\sum_{t=1}^{T} \langle \pi_t(\mathcal{A}_t) - \pi(\mathcal{A}_t), \theta_t\rangle\right] = \mathbb{E}\left[\sum_{t=1}^{T} \langle \Psi(\pi_t) - \Psi(\pi), \theta_t\rangle\right].$$

## 3.1 Approximating $\Omega$

Since $\Omega$ cannot be accessed directly without full knowledge of the context distribution $\mathcal{D}$, we cannot work with $\Omega$ directly. Instead, we will therefore approximate $\Omega$ by its empirical counterpart $\widehat{\Omega}$ based on $N$ i.i.d. samples $\mathcal{A}_1, \ldots, \mathcal{A}_N$ from $\mathcal{D}$:

$$
\widehat{\Omega} = \left\{ \hat{\Psi}(\pi) \Big| \pi \in \Pi \right\} = \left\{ x \in \mathbb{R}^d : x = \frac{1}{N} \sum_{i=1}^{N} a_i \ \middle| \ a_i \in \mathrm{conv}(\mathcal{A}_i), a_i = a_j \text{ when } \mathcal{A}_i = \mathcal{A}_j \right\},
\tag{3}
$$

where $\hat{\Psi}(\pi) = \frac{1}{N} \sum_{i=1}^{N} \pi(\mathcal{A}_i)$.

We proceed to show that the empirical cumulative loss of any linear classifier policy on the sample $\mathcal{A}_1, \ldots, \mathcal{A}_N$ is close to its expected cumulative loss, as long as $N$ is large enough:

**Lemma 1** (Uniform Convergence)**.** *Consider any loss vector $\theta \in \mathbb{R}^d$, and suppose that $|\mathcal{A}| \leq K$ and $\max_{a \in \mathcal{A}} |\langle a, \theta \rangle| \leq b$ almost surely. Then, for any $\delta \in (0, 1]$, uniformly over all linear classifier policies $\pi_\phi$, the difference in performance of $\pi_\phi$ on the sample $\mathcal{A}_1, \ldots, \mathcal{A}_N$ and its expected performance is at most*

$$
\sup_{\phi \in \mathbb{R}^d} \left| \langle \Psi(\pi_\phi), \theta \rangle - \langle \hat{\Psi}(\pi_\phi), \theta \rangle \right| \leq 2b \sqrt{\frac{2d \ln(N K^2)}{N}} + b \sqrt{\frac{2 \ln(4/\delta)}{N}}
$$

*with probability at least $1 - \delta$.*

The proof is provided in Appendix C. Its key idea is to rephrase the result as an equivalent statement about uniform convergence for linear multiclass classifiers with $K$ classes in the batch setting, with an unusual loss function. We can then obtain a concentration inequality that holds uniformly over all linear classifiers using standard tools. Specifically, we go via Rademacher complexity and a bound on the growth function of the class of multiclass linear classifiers in terms of its Natarajan dimension, which is known to be at most $d$.

## 3.2 Connection between Linear Contextual Bandits and Linear Bandits

---

**Algorithm 1:** Adversarial Linear Contextual Bandits

---

1 **Input**: An adversarial linear bandit algorithm ALG and a set $\mathcal{S}_N$ of $N$ action sets drawn from $\mathcal{D}$.
2 Initiate an instance of ALG over action set $\widehat{\Omega}$ constructed from $\mathcal{S}_N$.
3 **for** $t = 1, \ldots, T$ **do**
4      Obtain $y_t$ from ALG.
5      Find distribution $\alpha_t \in \Delta(\mathrm{vert}(\widehat{\Omega}))$ such that $\mathbb{E}_{\psi \sim \alpha_t}[\psi] = y_t$.
6      Sample $\psi_t \sim \alpha_t$ and let $\phi_t$ be an arbitrary element in the interior of $-\mathcal{N}(\widehat{\Omega}, \psi_t)$.
7      Receive action set $\mathcal{A}_t$, choose action $a_t = \arg\min_{a \in \mathcal{A}_t} \langle a, \phi_t \rangle$, and receive loss $\ell_t \in [0, 1]$.
8      Send $\ell_t$ to ALG.

---

The procedure that reduces linear contextual bandits to linear bandits is outlined in Algorithm 1. We also provide a visual illustration of $\widehat{\Omega}$ and the main variables in the algorithm in Figure 1. In the following discussion, let us assume for simplicity that the learner has perfect knowledge of $\Omega$, i.e., assume for now that $\widehat{\Omega} = \Omega$. Then, according to Section 3, we can view the problem as linear bandits over $\Omega \subset \mathbb{R}^d$. But when the linear bandit algorithm tells us to sample a point $y_t \in \Omega$, what corresponding policy $\pi_t$ should we use in order to guarantee that $\Psi(\pi_t) = \mathbb{E}_{\mathcal{A} \sim \mathcal{D}}[\pi_t(\mathcal{A})] = y_t$? This might be theoretically achievable if we knew $\mathcal{D}$ exactly, but even then the support of $\mathcal{D}$ might be too large to make this computationally tractable. To resolve this, we make a key observation stated in the following Lemma 2. In Lemma 2, we assume that $\mathcal{D}$ has a finite support and thus $\Omega$ is a polytope with a finite number of vertices. However, we do not really need this assumption in our setting, as we will eventually apply Lemma 2 on the empirical distribution $\widetilde{\mathcal{D}} = \mathrm{Uniform}\{\mathcal{A}_1, \ldots, \mathcal{A}_N\}$, which naturally has a finite support, and the set $\widehat{\Omega}$ built on it according to (3). The error between the empirical distribution $\widetilde{\mathcal{D}}$ and the true distribution $\mathcal{D}$ will be analyzed in Section 3.3.

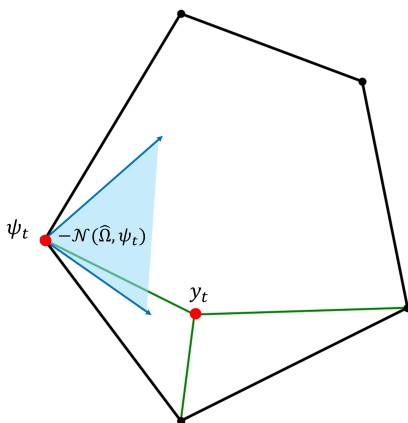

Figure 1: Illustration of $\widehat{\Omega}$, $y_t$, $\psi_t$ and the normal cone $-\mathcal{N}(\widehat{\Omega}, \psi_t)$.

**Lemma 2.** *Fix any finite-support action-set distribution $\mathcal{D}$ and its corresponding linear bandit feasible set $\Omega = \{\mathbb{E}_{\mathcal{A}\sim\mathcal{D}}[\pi(\mathcal{A})] : \text{ all possible policies } \pi\}$. For any vertex $\psi$ of $\Omega$, we can find a linear classifier policy $\pi_\phi$ such that $\mathbb{E}_{\mathcal{A}\sim\mathcal{D}}[\pi_\phi(\mathcal{A})] = \psi$. In fact, this holds for any interior point $\phi$ of $-\mathcal{N}(\Omega, \psi)$.*

*Proof of Lemma 2.* Let $\psi$ be a vertex of $\Omega$ and $\phi$ be an interior point of $-\mathcal{N}(\Omega, \psi)$. Then there exists $\phi$ such that for any $\psi' \in \Omega$, $\psi' \neq \psi$, $\langle \psi' - \psi, \phi \rangle > 0$.

Let $\tilde{\psi} = \mathbb{E}_{\mathcal{A}\sim\mathcal{D}}[\pi_\phi(\mathcal{A})]$, and let $\pi$ be a policy that $\psi$ corresponds to under distribution $\mathcal{D}$ (i.e., $\mathbb{E}_{\mathcal{A}\sim\mathcal{D}}[\pi(\mathcal{A})] = \psi$). Then

$$\left\langle \tilde{\psi}, \phi \right\rangle = \mathbb{E}_{\mathcal{A}\sim\mathcal{D}}\left[\langle \pi_\phi(\mathcal{A}), \phi \rangle\right] = \mathbb{E}_{\mathcal{A}\sim\mathcal{D}}\left[\min_{a\in\mathcal{A}}\left(a^\top \phi\right)\right] \leq \mathbb{E}_{\mathcal{A}\sim\mathcal{D}}\left[\pi(\mathcal{A})^\top \phi\right] = \langle \psi, \phi \rangle.$$

Thus, it must be that $\tilde{\psi} = \psi$. This finishes the proof. $\qquad\square$

Based on Lemma 2, we have the following strategy to *execute* $y_t$. First, decompose $y_t$ as a convex combination of extreme points of $\Omega$, i.e., decompose $y_t$ as $\sum_i \alpha_i \psi_i$ where $\psi_1, \psi_2, \ldots$ are vertices of $\Omega$, and $(\alpha_1, \alpha_2 \ldots)$ is a distribution over them. Then for each $\psi_i$, find the corresponding $\phi_i \in -\mathcal{N}(\Omega, \psi)$ according to Lemma 2. Finally, let $\pi_t$ be the policy that mixes $\pi_{\phi_1}, \pi_{\phi_2}, \ldots$ with weights $(\alpha_1, \alpha_2, \cdots)$. This way, we have by Lemma 2,

$$\mathbb{E}_{\mathcal{A}\sim\mathcal{D}}\left[\pi_t(\mathcal{A})\right] = \mathbb{E}_{\mathcal{A}\sim\mathcal{D}}\left[\sum_i \alpha_i \pi_{\phi_i}(\mathcal{A})\right] = \sum_i \alpha_i \mathbb{E}_{\mathcal{A}\sim\mathcal{D}}\left[\pi_{\phi_i}(\mathcal{A})\right] = \sum_i \alpha_i \psi_i = y_t.$$

This allows us to execute a policy $\pi_t$ such that $\mathbb{E}_{\mathcal{A}\sim\mathcal{D}}[\pi_t(\mathcal{A})] = y_t$ without explicit knowledge of $\mathcal{D} \in \Delta(2^{\mathbb{R}^d})$ but only knowledge of $\Omega \subset \mathbb{R}^d$. Notice that we do not have perfect knowledge of $\Omega$, but only the estimated feasible set $\widehat{\Omega}$. However, exactly the same approach can be applied to $\widehat{\Omega}$, resulting in the design of Algorithm 1.

### 3.3 Bounding the error due to the discrepancy between $\widehat{\Omega}$ and $\Omega$

From Section 3.2, we know how to *execute* the linear contextual bandit algorithm by leveraging a linear bandit procedure in $\widehat{\Omega}$. However, there are errors due to the discrepancy between $\widehat{\Omega}$ and $\Omega$ which contribute to the final regret.

First, the loss estimator we constructed will be *biased*. Suppose that $\widetilde{\mathcal{D}}$ is the empirical distribution based on $N$ action sets that are drawn independently from $\mathcal{D}$, and $\widehat{\Omega}$ is constructed from them based on (3). When we sample a point $y_t \in \widehat{\Omega}$ and execute the corresponding policy $\pi_t$ such that $y_t = \mathbb{E}_{\mathcal{A}\sim\widetilde{\mathcal{D}}}[\pi_t(\mathcal{A})]$, the expected loss the learner observes is $\mathbb{E}_{\mathcal{A}\sim\mathcal{D}}[\langle \pi_t(\mathcal{A}), \theta_t \rangle] \neq \langle y_t, \theta_t \rangle$. This

means that from the viewpoint of the linear bandit problem on $\widehat{\Omega}$, the feedback is *misspecified*. This motivates us to develop a misspecification robust linear bandit algorithm (Definition 1) which allows the feedback to not fully follow the standard linear bandit protocol. We elaborate more about this in Section 3.4.

The second source of error comes from difference between the action sets $\widehat{\Omega}$ and $\Omega$. With the misspecification-robust linear bandit algorithm on $\widehat{\Omega}$, the learner has good regret bound on $\widehat{\Omega}$. However, the real regret we care about is on $\Omega$. This requires us to bound the difference between the two regret definitions:

$$\text{Real regret on } \Omega \text{ that we care about: } \mathbb{E}_{\mathcal{A}\sim\mathcal{D}}[\langle \pi_t(\mathcal{A}), \theta_t \rangle] - \mathbb{E}_{\mathcal{A}\sim\mathcal{D}}[\langle \pi(\mathcal{A}), \theta_t \rangle],$$

Regret on $\widehat{\Omega}$ that the robust linear bandit algorithm can bound: $\mathbb{E}_{\mathcal{A}\sim\widetilde{\mathcal{D}}}[\langle \pi_t(\mathcal{A}), \theta_t \rangle] - \mathbb{E}_{\mathcal{A}\sim\widetilde{\mathcal{D}}}[\langle \pi(\mathcal{A}), \theta_t \rangle].$

Both errors discussed above can be related to the difference $\sup_\pi \left| \mathbb{E}_{\mathcal{A}\sim\widetilde{\mathcal{D}}}[\langle \pi(\mathcal{A}), \theta_t \rangle] - \mathbb{E}_{\mathcal{A}\sim\mathcal{D}}[\langle \pi(\mathcal{A}), \theta_t \rangle] \right|$. This can further be bounded using Lemma 1, where we establish uniform convergence over the set of all linear policies. According to Lemma 1, we have with probability at least $1 - \delta$, for all linear policies $\pi$,

$$\left| \mathbb{E}_{\mathcal{A}\sim\widetilde{\mathcal{D}}}[\langle \pi(\mathcal{A}), \theta_t \rangle] - \mathbb{E}_{\mathcal{A}\sim\mathcal{D}}[\langle \pi(\mathcal{A}), \theta_t \rangle] \right| \lesssim \sqrt{\frac{d\log(NK/\delta)}{N}}. \tag{4}$$

This allows us to bound the two sources of errors mentioned above.

### 3.4 Robust Linear Contextual Bandits

As discussed in Section 3.3, we would like to develop a linear bandit algorithm that tolerates misspecification. Although there is a rich related literature, most prior results are for the stochastic linear bandit problem and do not apply here. For adversarial linear bandits with misspecification robustness, we are only aware of the work by Neu and Olkhovskaya (2020) and Liu et al. (2024a). However, the bound in Neu and Olkhovskaya (2020) has a worse $T^{2/3}$ regret, while the algorithms of Liu et al. (2024a) are either computationally inefficient or highly sub-optimal. Fortunately, our problem is slightly easier than that studied by Liu et al. (2024a), as our learner has knowledge of the amount of misspecification $\epsilon$, and this amount remains the same in all rounds. This allows us to design the computationally efficient Algorithm 2 with an improved dependence on the amount of misspecification.

Algorithm 2 is an adaptation of the clipped continuous exponential weight algorithm of Ito et al. (2020): The $q_t$ in Line 3 of Algorithm 2 is the standard continuous exponential weights, while the $\widehat{q}_t$ in Line 5 confines the support of $q_t$ within an ellipsoid centered around the mean. This is helpful in obtaining a first-order bound (Ito et al., 2020). Sampling from $\widehat{q}_t$ can be done with standard techniques for sampling from a log-concave distribution (to sample from $q_t$) plus rejection sampling (to correct the distribution to $\widehat{q}_t$), which admits a $\text{poly}(d, C, T)$ computational complexity, provided that we have access to an efficient linear optimization oracle for $\hat{\Omega}$. See Ito et al. (2020)'s Section 4.4 for a discussion on the computational complexity. We discuss how to construct such an efficient linear optimization oracle with $\text{poly}(d, C, T)$ computational complexity in Section 4.

The key addition compared to (Ito et al., 2020) is the bonus term $b_t$ that ensures misspecification robustness. This bonus encourages additional exploration, preventing the learner from being misled by misspecified feedback. The form of the bonus for adversarial linear bandits was first developed in the series of work (Lee et al., 2020; Zimmert and Lattimore, 2022) aimed at obtaining high-probability bounds. Our use of the bonus is similar to Liu et al. (2024b), which tackles corruption and misspecification. In the regret analysis, this bonus term creates a negative regret term that offsets the regret overhead due to misspecification.

The guarantee of Algorithm 2 is given in the following theorem.

**Theorem 2.** *Algorithm 2 is a $\sqrt{d}$-misspecification-robust linear bandit algorithm defined in Definition 1.*

We remark that while there exist algorithms that are 1-misspecification-robust (Liu et al., 2024a), their run time scales at least with the number of actions. Algorithm 2 achieves $\alpha$-robustness with the smallest $\alpha$ we are aware of among algorithms that run in $\text{poly}(d, C, T)$ time.

---

**Algorithm 2:** Misspecification-Robust Continuous Exponential Weights

---
1 **Parameters**: $\gamma = 10 \log(10dT)$, $\beta = T^{-4}$.
2 **for** $t = 1, 2, \ldots, T$ **do**
3     Define

$$
q_t(y) = \frac{\exp\left(-\eta \sum_{\tau < t} \left\langle y, \widehat{\theta}_\tau - b_\tau \right\rangle\right)}{\int_{\widehat{\Omega}} \exp\left(-\eta \sum_{\tau < t} \left\langle z, \widehat{\theta}_\tau - b_\tau \right\rangle\right) \mathrm{d}z}, \quad x_t = \mathbb{E}_{y \sim q_t}[y], \quad \Sigma_t = \mathbb{E}_{y \sim q_t}[(y - x_t)(y - x_t)^\top].
$$

5     Define

$$
\widehat{q}_t(y) = \frac{q_t(y) \mathbb{I}\left\{\|y - x_t\|^2_{\Sigma_t^{-1}} \le d\gamma^2\right\}}{\int_{\widehat{\Omega}} q_t(z) \mathbb{I}\left\{\|z - x_t\|^2_{\Sigma_t^{-1}} \le d\gamma^2\right\} \mathrm{d}z}, \quad \widehat{\Sigma}_t = \mathbb{E}_{y \sim \widehat{q}_t}[(y - x_t)(y - x_t)^\top].
$$

6     Sample $y_t \sim \widehat{q}_t$ and receive loss $c_t \in [0, 1]$.
7     Define $\widehat{\theta}_t = (\beta I + \widehat{\Sigma}_t)^{-1}(y_t - x_t)c_t$ and $b_t = 8\eta \left(\epsilon + \frac{1}{T^2}\right) \sum_{\tau < t}(\widehat{\theta}_\tau - b_\tau)$.

---

In fact, Algorithm 2 achieves an even more favorable small-loss regret bound, which can be leveraged to obtain a small-loss bound for linear contextual bandits when the simulator is available. We discuss this in Section 5.

### 3.5 Combining Everything and Using the Doubling Trick

Combining everything above, we are able to establish the regret bound for the linear contextual bandit problem. The proof of the following theorem is in Appendix D.2.

**Theorem 3.** *Algorithm 1 with* ALG *instantiated as a $\alpha$-misspecification-robust linear bandit algorithm ensures*

$$
\mathbb{E}\left[\sum_{t=1}^T \langle a_t, \theta_t \rangle\right] \le \min_{\pi \in \Pi} \mathbb{E}\left[\sum_{t=1}^T \langle \pi(\mathcal{A}_t), \theta_t \rangle\right] + \tilde{O}\left(d\sqrt{T} + \alpha T d \sqrt{\frac{\log(NKT)}{N}}\right).
$$

**Corollary 1** (Restatement of Theorem 1). *Given access to a $\alpha$-misspecification-robust adversarial linear bandit algorithm* ALG*, Algorithm 1 with doubling trick achieves* $\max_{\pi \in \Pi} \mathrm{Reg}_T(\pi) \le \tilde{O}(d\sqrt{T} + \alpha d\sqrt{T \log K})$ *in adversarial linear contextual bandits without access to simulators.*

*Proof.* We will use the doubling trick and restart Algorithm 1 at times $2, 4, 8, 16, \ldots$, each time using the contexts received so far to estimate $\widehat{\Omega}$. Thus, in the $k$-th epoch, $\widehat{\Omega}$ is constructed by $N = \Theta(2^k)$ contexts, allowing us to bound the regret in epoch $k$ as

$$
\tilde{O}\left(d\sqrt{2^k} + \alpha 2^k d \sqrt{\frac{\log(NKT)}{2^k}}\right) = \tilde{O}\left(d\sqrt{2^k} + \alpha d \sqrt{2^k \log(NKT)}\right)
$$

using Theorem 3. Summing the regret over all epochs allows us to bound

$$
\max_{\pi \in \Pi} \mathrm{Reg}_T(\pi) \le \tilde{O}\left(\sum_{k=1}^{\log_2 T} \left(d\sqrt{2^k} + \alpha d \sqrt{2^k \log(NKT)}\right)\right) = \tilde{O}\left(d\sqrt{T} + \alpha d \sqrt{T \log K}\right).
$$

$\square$

By instantiating ALG as Algorithm 2 (which is an $\sqrt{d}$-misspecification-robust algorithm by Theorem 2) and invoking Corollary 1, we get the final regret bound as $\tilde{O}(\sqrt{d^3 T \log K})$. As we can assume $K \le T^d$ without loss of generality (any action set can be discretized into no more than $T^d$ points and incurs a negligible regret of $d/T$ due to discretization error), the regret bound can be further improved to $\tilde{O}(\min\{d^2\sqrt{T}, \sqrt{d^3 T \log K}\})$.

## 4  Computational Complexity

In this section we discuss how to construct the $\text{poly}(d, C, N)$ time linear optimization oracle needed in Section 3.4 to make Algorithm 2 efficient; we also consider two steps from Algorithm 1, lines 5 and 6, for which it is not obvious how they can be implemented efficiently. We will describe an approach to implement both steps in $\text{poly}(d, C, N)$ time, provided that $\text{conv}(\mathcal{A})$ is a polytope that can be described by at most $C$ linear constraints for $\mathcal{D}$-almost all $\mathcal{A}$.

**Constructing Linear Optimization and Separation Oracles.**  By Corollary 14.1a of Schrijver (1986) (restated in Lemma 4), we can obtain a $\text{poly}(d, C, N)$ time linear optimization oracle for $\widehat{\Omega}$ from a $\text{poly}(d, C, N)$ time separation oracle for $\widehat{\Omega}$. We proceed to construct such a separation oracle for $\widehat{\Omega}$. To this end, let us assume, without loss of generality, that no two sets $\mathcal{A}_i$ and $\mathcal{A}_j$ in the construction of $\widehat{\Omega}$ are equal; otherwise we can replace them by a single set $2\mathcal{A}_i$, which will only decrease $N$. Thus, $\widehat{\Omega}$ is the Minkowski sum of $N$ convex polytopes that, by assumption, can all be described by at most $C$ constraints. For any $x \in \mathbb{R}^d$, let

$$g(\phi) = \phi^\top x - \max_{x' \in \widehat{\Omega}} \phi^\top x' = \phi^\top x - \tfrac{1}{N} \sum_{i=1}^{N} \max_{x_i \in \mathcal{A}_i} \phi^\top x_i.$$

If $g(\phi) > 0$ for some $\phi$, then $\phi$ gives a hyperplane that separates $x$ from $\widehat{\Omega}$; and if $g(\phi) \leq 0$ for all $\phi$, then $x \in \widehat{\Omega}$. Since $g$ is concave, and we can solve every subproblem $\max_{x_i \in \mathcal{A}_i} \phi^\top x_i$ in $\text{poly}(d, C)$ time, we can maximize $g$ in $\text{poly}(d, C, N)$ time to obtain our separation oracle.

**Implementing line 5 of Algorithm 1.**  To implement line 5, we need to be able to take any point $y_t \in \widehat{\Omega}$ and find a distribution $\alpha_t$ supported on the vertices of $\widehat{\Omega}$ that we can sample from efficiently. By Carathéodory's theorem, any $y_t \in \widehat{\Omega}$ can be represented as a convex combination of at most $d + 1$ vertices of $\widehat{\Omega}$: $y_t = \sum_{l=1}^{k} \alpha_l v_l$ for $k \leq d + 1$ where all $v_l$ are vertices of $\widehat{\Omega}$, and $\alpha_1, \ldots, \alpha_k \geq 0$ with $\sum_l \alpha_l = 1$ can be interpreted as the probabilities of selecting the vertices. This is a categorical distribution on at most $d + 1$ points, from which we can sample in $O(d)$ time (assuming we have access to an oracle that provides samples from the uniform distribution on $[0, 1]$). Thus the main challenge is to compute the vertices $v_l$ and the probabilities $\alpha_l$. By Corollary 14.1g) of Schrijver (1986) (restated in Lemma 3), there exists an algorithm that can do both in time $\text{poly}(d, h)$ for any set $\widehat{\Omega}$ given access to a separation oracle that runs in time $O(h)$. As discussed above, we have $h$ of order $\text{poly}(d, C, N)$, from which it follows that we can also find the probabilities $\alpha_l$ and vertices $v_l$ in $\text{poly}(d, C, N)$ time.

**Implementing line 6 of Algorithm 1.**  After the sampling procedure has chosen a particular vertex $\psi_t \in \{v_1, \ldots, v_k\}$, we need to find an interior point of the corresponding normal cone $-\mathcal{N}(\widehat{\Omega}, \psi_t)$ in line 6. To achieve this, suppose we are given a vertex $\psi_t \in \widehat{\Omega}$. By Lemma 5, there exists a unique decomposition $\psi_t = \frac{1}{N} \sum_{i=1}^{N} x_i$ where $x_i$ is a vertex of $\text{conv}(\mathcal{A}_i)$ for all $i \in [N]$. We can find $\{x_i\}_{i \in [N]}$ by solving a linear program with $O(NC)$ constraints: $x_i \in \text{conv}(\mathcal{A}_i)$ for all $i \in [N]$ and $\frac{1}{N} \sum_{i=1}^{N} x_i = \psi_t$.

Let $\mathcal{N}_i = \mathcal{N}(\text{conv}(\mathcal{A}_i), x_i)$ be the normal cone of $\text{conv}(\mathcal{A}_i)$ at the vertex $x_i$. It is the conic hull of the active constraint normals: $\mathcal{N}_i = \text{cone}(\mathcal{T}_i) = \left\{ \sum_{u \in \mathcal{T}_i} \lambda_{i,u} u \mid \lambda_{i,u} \geq 0 \text{ for all } u \in \mathcal{T}_i \right\}$, where $\mathcal{T}_i$ is the set of normal vectors $u$ corresponding to the constraints $u^\top x + v \leq 0$ that are active at $x_i$. $\mathcal{T}_i$ can be constructed by enumerating all constraints of $\text{conv}(\mathcal{A}_i)$ and checking which are tight at $x_i$.

By Lemma 6, $\mathcal{N}(\widehat{\Omega}, \psi_t) = \bigcap_{i \in [N]} \mathcal{N}_i$. To find a point strictly inside $-\mathcal{N}(\widehat{\Omega}, \psi_t)$, we solve the following linear program with $O(NC)$ constraints:

$$\max_{\varepsilon \in \mathbb{R},\ \lambda_{i,u} \in \mathbb{R},\ \phi \in \mathbb{R}^d} \varepsilon$$

$$\text{subject to } \phi = -\sum_{u \in \mathcal{T}_i} \lambda_{i,u} u \quad \text{for all } i \in [N], \hspace{3cm} (\phi \in -\mathcal{N}_i \text{ for all } i)$$

$$\varepsilon \leq \lambda_{i,u} \leq 1 \quad \text{for all } i \in [N] \text{ and } u \in \mathcal{T}_i. \ \ (\text{Make } \phi \text{ bounded and strictly inside } -\mathcal{N}_i)$$

By the assumption that $\psi_t$ is a vertex, $-\mathcal{N}(\widehat{\Omega}, \psi_t)$ always has an non-empty interior. Therefore, a solution with $\varepsilon > 0$ exists, and the optimal $\phi$ provides a desired interior point.

# 5   Small-Loss Bound with Access to Simulator

The sub-optimal rate $d^2\sqrt{T}$ we obtained in Section 3 comes from the misspecification term $\alpha\sqrt{d}\epsilon T$ in the regret bound of robust linear bandits (Definition 1). While it is unclear how to further improve $\alpha$ or $\epsilon$, we demonstrate the power of our reduction by further assuming access to simulator: it not only allows us to recover the minimax optimal regret $d\sqrt{T}$ but also allows us to obtain a first-order bound $d\sqrt{L^\star}$ when losses are non-negative, where $L^\star$ is the cumulative loss of the best policy.

By the black-box nature of our reduction, what we additionally need is just a misspecification-robust linear bandit algorithm with *small-loss* regret bound guarantee, formally defined as follows:

**Definition 2** ($\alpha$-misspecification-robust adversarial linear bandit algorithm with small-loss bounds)**.** *A misspecification-robust linear bandit algorithm with small-loss bounds over action set $\widehat{\Omega} \subset \mathbb{R}^d$ has the following property: with a given $\epsilon > 0$ and the guarantee that every time the learner chooses $y_t \in \widehat{\Omega}$, the loss received $c_t \in [0,1]$ satisfies $|\mathbb{E}_t[c_t] - \langle y_t, \theta_t\rangle| \leq \epsilon$, the algorithm ensures*

$$\mathbb{E}\left[\sum_{t=1}^{T}\langle y_t, \theta_t\rangle\right] \leq \min_{y\in\widehat{\Omega}}\sum_{t=1}^{T}\langle y, \theta_t\rangle + \widetilde{O}\left(d\sqrt{\sum_{t=1}^{T}\langle y, \theta_t\rangle} + \alpha\sqrt{d}\epsilon T\right). \tag{5}$$

The next theorem shows that Algorithm 2 satisfies Definition 2 with $\alpha = \sqrt{d}$.

**Theorem 4.** *Algorithm 2 is a $\sqrt{d}$-misspecification-robust linear bandit algorithm with small-loss bound defined in Definition 2.*

With access to a misspecification-robust linear bandit algorithm with small-loss bounds, we have

**Theorem 5.** *Given access to simulator that can generate free contexts, and access to an $\alpha$-misspecification-robust adversarial linear bandit algorithm with small-loss regret bound guarantee, we can achieve $\min_{\pi\in\Pi}\mathbb{E}\left[\mathrm{Reg}_T(\pi)\right] \leq \widetilde{O}(d\sqrt{L^\star})$ in adversarial linear contextual bandits, where*

$$L^\star = \min_{\pi\in\Pi}\mathbb{E}\left[\sum_{t=1}^{T}\langle\pi(\mathcal{A}_t), \theta_t\rangle\right]$$

*is the expected total loss of the best policy. This is achieved with $O(\alpha^2 d^2 T^2)$ calls to the simulator.*

The proof of Theorem 5 is very similar to Theorem 3, except that now, with access to the simulator, we are able to make $N$ in Theorem 3 large enough that the second term in Theorem 3 is negligible. We provide the omitted proofs in Appendix E.

# 6   Conclusion and Open Questions

We have provided a general framework that reduces adversarial linear contextual bandits to misspecification-robust linear bandits in a black-box manner. It achieves $\widetilde{\mathcal{O}}(d^2\sqrt{T})$ regret without a simulator, and is the first algorithm we know of that handles combinatorial bandits with stochastic action sets and adversarial losses efficiently. The requirement of misspecification robustness stems from our need to use an approximate feasible set $\widehat{\Omega}$ because we do not have direct access to the exact feasible set $\Omega$, which depends on the action set distribution $\mathcal{D}$.

Three open questions are left by our work: First, can the computation cost be improved further to match that of Neu and Valko (2014), who only require a linear optimization oracle for each action set, and do not require a polynomial number of constraints? Second, can the regret be further improved to the near-optimal $\widetilde{\mathcal{O}}(d\sqrt{T})$ bound with a polynomial-time algorithm without simulators? Third, can we achieve $\mathrm{poly}(d)\sqrt{L^\star}$ regret without simulator? For the second question, one may try to generalize the approach of Schneider and Zimmert (2023). For the third question, one idea is to establish a Bernstein-type counterpart of Lemma 1, potentially drawing ideas from Bartlett et al. (2005); Liang et al. (2015).

We expect that our approach has wider applications than adversarial linear contextual bandits. For example, our approach may be generalized to linear MDPs with fixed transition and adversarial losses (Luo et al., 2021; Sherman et al., 2023; Dai et al., 2023; Kong et al., 2023; Liu et al., 2024b) and facilitate learning with exponentially large or continuous action sets.

## Acknowledgments and Disclosure of Funding

Mayo and Van Erven were supported by the Netherlands Organization for Scientific Research (NWO) under grant number VI.Vidi.192.095.

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

## A  Comparison with Hanna et al. (2023)

Our work and Hanna et al. (2023) both reduce linear contextual bandits to linear bandits with misspecification. The linear bandits the two works reduce to are slightly different. For comparison, let $\mathcal{A}_1, \ldots, \mathcal{A}_N$ be empirical action sets drawn from $\mathcal{D}$, and define

$$\widehat{\Omega} = \left\{ \hat{\Psi}(\pi) \Big| \pi \in \Pi \right\}, \qquad \widehat{\Omega}_{\text{lin}} = \left\{ \hat{\Psi}(\pi) \Big| \pi \in \Pi_{\text{lin}} \right\}, \qquad \widehat{\Omega}_{\text{lin}}^{\epsilon} = \left\{ \hat{\Psi}(\pi) \Big| \pi \in \Pi_{\text{lin}}^{\epsilon} \right\}, \qquad (6)$$

where $\hat{\Psi}(\pi) = \frac{1}{N} \sum_{i=1}^{N} \pi(\mathcal{A}_i)$, $\Pi_{\text{lin}}$ is the set of linear policies defined in (1), and

$$\Pi_{\text{lin}}^{\epsilon} = \left\{ \pi_\phi \mid \phi \in \epsilon\text{-net of the unit ball in } \mathbb{R}^d \right\}.$$

It holds that $\widehat{\Omega}_{\text{lin}}^{\epsilon} \subset \widehat{\Omega}_{\text{lin}} \subset \widehat{\Omega}$ because $\Pi_{\text{lin}}^{\epsilon} \subset \Pi_{\text{lin}} \subset \Pi$. The relation among the three sets in (6) is the following: $\widehat{\Omega}_{\text{lin}}$ is the boundary of $\widehat{\Omega}$ (Lemma 2), and $\widehat{\Omega}_{\text{lin}}^{\epsilon}$ is a discretized subset of $\widehat{\Omega}_{\text{lin}}$. It has been shown that there must exist an optimal policy in $\Pi_{\text{lin}}$ (see our arguments in Section 2.1), and that $\Pi_{\text{lin}}^{\epsilon}$ must contain an $\epsilon$-near-optimal policy (see (47) of Liu et al. (2023)).

Our work reduces the linear contextual bandit problem to linear bandits in $\widehat{\Omega}$, while Hanna et al. (2023) reduces it to linear bandits in $\widehat{\Omega}_{\text{lin}}^{\epsilon}$. According to the discussion above, both suffice to establish no regret guarantees against the optimal policy. The key difference lies in the computational efficiency. As discussed in Section 4, $\widehat{\Omega}$ is a polytope whose separation oracle can be implemented in $\text{poly}(d, C, T)$ time, while in Hanna et al. (2023), $\widehat{\Omega}_{\text{lin}}^{\epsilon}$ is a set containing $(\frac{1}{\epsilon})^{\Theta(d)}$ discrete points with $\epsilon$ chosen as $\Theta(\frac{1}{T})$. It remains open whether $\widehat{\Omega}_{\text{lin}}^{\epsilon}$ can be represented efficiently and whether Hanna et al. (2023)'s algorithm can be implemented in polynomial time. In addition, our work operates on the adversarial-loss setting, strictly generalizing Hanna et al. (2023)'s stochastic-loss setting.

A technical question related to the discussion above is whether we can apply the discretization idea from Hanna et al. (2023) purely in the analysis but not in the algorithm, i.e., bounding the regret overhead by the discretization error without restricting the learner's policy to the discretized linear policies. This would allow us to avoid the Rademacher complexity analysis (Lemma 1) and instead use Hoeffding's or Bernstein's inequality combined with a union bound, possibly leading to an improved regret bound. However, there is a key technical challenge: the mapping $\hat{\Psi}(\pi_\phi)$ is not continuous in $\phi$, which prevents us from bounding $\|\hat{\Psi}(\pi_\phi) - \hat{\Psi}(\pi_{\phi'})\|$ by a constant times $\|\phi - \phi'\|$, where $\phi$ is an arbitrary point and $\phi'$ is a point in the discretized set. This would leave us no control on the performance of policies outside the set of discretized linear policies $\Pi_{\text{lin}}^{\epsilon}$. As a result, we are unable to derive a uniform convergence result analogous to Lemma 1 using this approach.

## B  Lemmas for Linear Programming

**Lemma 3** (Corollaries 14.1f and 14.1g of Schrijver (1986)). *Suppose $P \subset \mathbb{R}^d$ is a bounded polytope defined by rational inequalities of size at most $\alpha$, and for which there exists a separation oracle* SEP. *There exists an algorithm that solves the following problem: for any rational $x \in P$, find vertices $x_0, x_1, \ldots, x_d \in P$ and $\lambda_0, \lambda_1, \ldots, \lambda_d \geq 0$ such that $x = \sum_{i=0}^{d} \lambda_i x_i$ and $\sum_{i=0}^{d} \lambda_i = 1$, in time polynomially bounded by $d$, $\alpha$, the running time of* SEP, *and the size of the input $x$.*

**Lemma 4** (Corollary 14.1a of Schrijver (1986)). *Suppose $P \subset \mathbb{R}^d$ is a polytope defined by rational inequalities of size at most $\alpha$, and for which there exists a separation oracle* SEP. *Then there exists an algorithm that solves the linear optimization problem $\arg\max_{x \in P} \beta^\top x$ for any rational vector $\beta$ in time polynomially bounded by $d$, $\alpha$, the running time of* SEP, *and the size of the input $\beta$.*

**Lemma 5** (Theorem 3.1.2 of Weibel (2007)). *Let $P_1, \ldots, P_N$ be polytopes in $\mathbb{R}^d$. Let $F$ be a face of the Minkowski sum $\sum_{i=1}^{N} P_i$. Then there are faces $F_1, F_2, \ldots, F_N$ of $P_1, P_2, \ldots, P_N$ respectively such that $F = \sum_{i=1}^{N} F_i$. What's more, this decomposition is unique.*

**Lemma 6** (Corollary 3.1.3 of Weibel (2007)). *Let $P = \sum_{i=1}^{N} P_i$ be a Minkowsky sum of polytopes in $\mathbb{R}^d$, let $F$ be a nonempty face of $P$, and let $F_1, \ldots, F_N$ be its decomposition. Then $\mathcal{N}(P, F) = \mathcal{N}(P_1, F_1) \cap \cdots \cap \mathcal{N}(P_N, F_N)$.*

## C  Proof of Lemma 1

**Lemma 1** (Uniform Convergence). *Consider any loss vector $\theta \in \mathbb{R}^d$, and suppose that $|\mathcal{A}| \leq K$ and $\max_{a \in \mathcal{A}} |\langle a, \theta \rangle| \leq b$ almost surely. Then, for any $\delta \in (0, 1]$, uniformly over all linear classifier policies $\pi_\phi$, the difference in performance of $\pi_\phi$ on the sample $\mathcal{A}_1, \ldots, \mathcal{A}_N$ and its expected performance is at most*

$$\sup_{\phi \in \mathbb{R}^d} \left| \langle \Psi(\pi_\phi), \theta \rangle - \langle \hat{\Psi}(\pi_\phi), \theta \rangle \right| \leq 2b\sqrt{\frac{2d \ln(NK^2)}{N}} + b\sqrt{\frac{2 \ln(4/\delta)}{N}}$$

*with probability at least $1 - \delta$.*

*Proof.* We will first reduce the result to a statement about uniform convergence for linear multiclass classifiers with $K$ classes and an unusual loss function. To this end, note first that we can assume without loss of generality that $\mathcal{A} = \{V_1, \ldots, V_K\}$, where the $V_1, \ldots, V_K$ are random vectors in $\mathbb{R}^d$. (If $\mathcal{A}$ has fewer than $K$ elements, then consider it as a multiset and add repetitions of one of its elements.) We can assume all randomness in $\mathcal{A}$ is determined by an underlying random variable $Z$ and that $V_y = -g(Z, y)$ for each 'class' $y \in [K]$, where $g$ is a class-sensitive feature map in the sense of Shalev-Shwartz and Ben-David (2014, Section 17.2). Defining the linear multiclass classifier

$$c_\phi(Z) = \underset{y \in [K]}{\arg\max} \langle g(Z, y), \phi \rangle,$$

we then obtain the correspondence

$$\pi_\phi(\mathcal{A}) = V_{c_\phi(Z)}.$$

For any multiclass classifier $c$, let $f(Z, c) = \langle V_{c(Z)}, \theta \rangle$ be our 'loss function'. Then

$$\langle \Psi(\pi_\phi), \theta \rangle = \mathbb{E}_Z[f(Z, c_\phi)], \qquad \langle \hat{\Psi}(\pi_\phi), \theta \rangle = \frac{1}{N} \sum_{i=1}^{N} f(Z_i, c_\phi).$$

We therefore need to show the following uniform convergence result, with probability at least $1 - \delta$,

$$\sup_{c \in \mathcal{C}} \left| \mathbb{E}[f(Z, c)] - \frac{1}{N} \sum_{i=1}^{N} f(Z_i, c) \right| \leq 2b\sqrt{\frac{2d \ln(NK^2)}{N}} + b\sqrt{\frac{2 \ln(4/\delta)}{N}} \qquad (7)$$

for the class of linear multiclass classifiers

$$\mathcal{C} = \Big\{ c_\phi \mid \phi \in \mathbb{R}^d \Big\},$$

with loss function $f$. In order to establish (7), let $S = (Z_1, \ldots, Z_N)$, and consider the empirical Rademacher complexity

$$\mathrm{Rad}(\mathcal{C}, S) = \frac{1}{N} \underset{\sigma_1, \ldots, \sigma_N}{\mathbb{E}} \left[ \sup_{c \in \mathcal{C}} \sum_{i=1}^{N} \sigma_i f(Z_i, c) \right],$$

where the $\sigma_i$ are independent Rademacher random variables with $\Pr(\sigma_i = -1) = \Pr(\sigma_i = +1) = 1/2$. Since $|f(Z, c)| \leq b$ for $c \in \mathcal{C}$ by assumption, standard concentration bounds in terms of Rademacher complexity imply that

$$\sup_{c \in \mathcal{C}} \left| \mathbb{E}_Z[f(Z, c)] - \frac{1}{N} \sum_{i=1}^{N} f(Z_i, c) \right| \leq 2 \underset{S'}{\mathbb{E}}[\mathrm{Rad}(\mathcal{C}, S')] + b\sqrt{\frac{2 \ln(4/\delta)}{N}} \qquad (8)$$

with probability at least $1 - \delta$. (This follows, for instance, by observing that

$$\sup_{c \in \mathcal{C}} |\mathbb{E}_Z[f(Z, c)] - \frac{1}{N} \sum_{i=1}^{N} f(Z_i, c)|$$

$$= \max \left\{ \sup_{c \in \mathcal{C}} \mathbb{E}_Z[f(Z, c)] - \frac{1}{N} \sum_{i=1}^{N} f(Z_i, c), \sup_{c \in \mathcal{C}} \mathbb{E}_Z[-f(Z, c)] - \frac{1}{N} \sum_{i=1}^{N} (-f(Z_i, c)) \right\}$$

and applying part 1 of Theorem 26.5 of Shalev-Shwartz and Ben-David (2014) separately to $f$ and $-f$ to control both parts in the maximum separately using a union bound; then noting that $f$ and $-f$ have the same Rademacher complexity.)

We proceed to bound the Rademacher complexity on the right-hand side of (8). First, let $\mathcal{C}_S = \{(c(Z_1), \ldots, c(Z_N)) \mid c \in \mathcal{C}\}$ denote the restriction of $\mathcal{C}$ to the sample $S$. Then

$$\text{Rad}(\mathcal{C}, S) = \text{Rad}(\mathcal{C}_S, S) \leq b\sqrt{\frac{2\ln|\mathcal{C}_S|}{N}}$$

by Massart's lemma (Shalev-Shwartz and Ben-David, 2014). As discussed by Shalev-Shwartz and Ben-David (2014, Chapter 29), one possible generalization of the Vapnik-Chervonenkis dimension to multiclass classification is Natarajan's dimension $\text{NatDim}(\mathcal{C})$. Natarajan's lemma (Natarajan, 1989, p. 93), (Shalev-Shwartz and Ben-David, 2014, Lemma 29.4) shows that

$$|\mathcal{C}_S| \leq N^{\text{NatDim}(\mathcal{C})} K^{2\,\text{NatDim}(\mathcal{C})},$$

and, for linear multiclass classifiers, it is also known (Shalev-Shwartz and Ben-David, 2014, Theorem 29.7) that

$$\text{NatDim}(\mathcal{C}) \leq d.$$

Putting all inequalities together, it follows that

$$\sup_{c \in \mathcal{C}} \left| \mathbb{E}_Z[f(Z, c)] - \frac{1}{N}\sum_{i=1}^{N} f(Z_i, c) \right| \leq 2b\sqrt{\frac{2d\ln(NK^2)}{N}} + b\sqrt{\frac{2\ln(4/\delta)}{N}}$$

with probability at least $1 - \delta$, as required. $\qquad\square$

# D   Omitted Details in Section 3

## D.1   Robust Linear Bandits

**Lemma 7** (Lemma 14 of Zimmert and Lattimore (2022))**.** *Let $F$ be a $\nu$-self-concordant barrier for $\mathcal{A} \subset \mathbb{R}^d$ for some $\nu \geq 1$. Then for any $x, u \in \mathcal{A}$,*

$$\|x - u\|_{\nabla^2 F(x)} \leq \gamma'\langle x - u, \nabla F(x)\rangle + 6\gamma'\nu$$

*where $\gamma' = \frac{8}{3\sqrt{3}} + \frac{7^{\frac{3}{2}}}{6\sqrt{3\nu}}$ ($\gamma' \in [1, 4]$ for $\nu \geq 1$).*

It is known that the continuous exponential weight algorithm is equivalent to FTRL with entropic barrier as the regularizer together with a particular sampling scheme (Bubeck and Eldan, 2015; Zimmert and Lattimore, 2022). We summarize the equivalence in the following lemma, the details of which can be found in Zimmert and Lattimore (2022).

**Lemma 8** (Facts from Bubeck and Eldan (2015), Zimmert and Lattimore (2022))**.** *Consider Algorithm 2. Let $x_t = \mathbb{E}_{y \sim q_t}[y]$ and let $F : \mathcal{A} \to \mathbb{R}$ be the entropic barrier on $\mathcal{A}$. Then we have*

$$x_t = \operatorname*{arg\,min}_{y \in \widehat{\Omega}} \left\langle y, \sum_{\tau < t}\left(\widehat{\theta}_\tau - b_\tau\right)\right\rangle + \frac{F(y)}{\eta}.$$

*Furthermore,*

$$\nabla F(x_t) = -\eta\sum_{\tau < t}(\widehat{\theta}_\tau - b_\tau) \quad and \quad \nabla^2 F(x_t) = \left(\mathbb{E}_{y \sim q_t}[(y - x_t)(y - x_t)^\top]\right)^{-1}.$$

**Lemma 9** (Lemma 4, Ito et al. (2020))**.** *Let $q_t, \widehat{q}_t, \Sigma_t, \widehat{\Sigma}_t$ be as defined in Algorithm 2. Then for any $f(y) : \mathcal{A} \to [-1, 1]$,*

$$|\mathbb{E}_{y \sim q_t}[f(y)] - \mathbb{E}_{y \sim \widehat{q}_t}[f(y)]| \leq 10d\exp(-\gamma) \leq \frac{1}{d^5 T^{10}}.$$

*Furthermore,*

$$\frac{3}{4}\Sigma_t \preceq \widehat{\Sigma}_t \preceq \frac{4}{3}\Sigma_t.$$

*Proof of Theorem 2.* First, we decompose the regret as follows:

$$\mathbb{E}\left[\sum_{t=1}^{T}\langle y_t - u, \theta_t\rangle\right]$$

$$\leq \mathbb{E}\left[\sum_{t=1}^{T}\langle \mathbb{E}_{y\sim q_t}[y] - u, \theta_t\rangle\right] + O(1)$$

$$= \underbrace{\mathbb{E}\left[\sum_{t=1}^{T}\left\langle x_t - u, \widehat{\theta}_t - b_t\right\rangle\right]}_{\textbf{FTRL}} + \underbrace{\mathbb{E}\left[\sum_{t=1}^{T}\left\langle x_t - u, \theta_t - \widehat{\theta}_t\right\rangle\right]}_{\textbf{Bias}} + \underbrace{\mathbb{E}\left[\sum_{t=1}^{T}\langle x_t - u, b_t\rangle\right]}_{\textbf{Bonus}} + O(1).$$

**Bounding FTRL term**  The analysis for the **FTRL** term follows that in Ito et al. (2020). Specifically, directly following their Lemma 5, Lemma 6, and the analysis below Lemma 6, we have the following: as long as $\eta\|\widehat{\theta}_t - b_t\|_{\Sigma_t} \leq 1$, we have

$$\textbf{FTRL} \leq \frac{d\log T}{\eta} + \eta\mathbb{E}\left[\sum_{t=1}^{T}\|\widehat{\theta}_t - b_t\|_{\Sigma_t}^2\right] + O(1)$$

$$\leq \frac{d\log T}{\eta} + 2\eta\mathbb{E}\left[\sum_{t=1}^{T}\|\widehat{\theta}_t\|_{\Sigma_t}^2 + \|b_t\|_{\nabla^{-2}F(x_t)}^2\right] + O(1). \qquad \text{(by Lemma 8)}$$

For the two middle terms, we have

$$\|\widehat{\theta}_t\|_{\Sigma_t}^2 \leq (y_t - x_t)^{\top}\left(\beta I + \widehat{\Sigma}_t\right)^{-1}\Sigma_t\left(\beta I + \widehat{\Sigma}_t\right)^{-1}(y_t - x_t)c_t^2$$

$$\leq 2(y_t - x_t)^{\top}\Sigma_t^{-1}(y_t - x_t)c_t^2 \qquad \text{(by Lemma 9)}$$

$$\leq 2d\gamma^2 c_t^2, \qquad \text{(by the truncation in the algorithm)}$$

and

$$\|b_t\|_{\nabla F(x_t)}^2 = \left(2\sqrt{d}\gamma\epsilon + \frac{1}{T^2}\right)^2\|\nabla F(x_t)\|_{\nabla^{-2}F(x_t)}^2 \leq O\left(d^2\gamma^2\epsilon^2 + \frac{1}{T^2}\right),$$

where we used the fact that $F(\cdot)$ is a $O(d)$ self-concordant barrier and thus $\|\nabla F(x_t)\|_{\nabla^{-2}F(x_t)}^2 \leq O(d)$. Thus, since $\eta \leq \frac{1}{d\gamma^2}$, we have

$$\textbf{FTRL} \leq \frac{d\log T}{\eta} + O\left(\eta d\gamma^2\sum_{t=1}^{T}c_t^2 + \eta T d^2\gamma^2\epsilon^2\right).$$

**Bounding Bias term**  Let $\mathbb{E}[c_t] = y_t^{\top}\theta_t + \epsilon_t(y_t)$, where $\epsilon_t(y)$ is the amount of misspecification when choosing $y$. By assumption, we have $|\epsilon_t(y)| \leq \epsilon$ for any $y$.

$$\mathbb{E}_t[\widehat{\theta}_t] = \mathbb{E}_t\left[(\beta I + \widehat{\Sigma}_t)^{-1}(y_t - x_t)\left(y_t^{\top}\theta_t + \epsilon_t(y_t)\right)\right]$$

$$= \mathbb{E}_t\left[(\beta I + \widehat{\Sigma}_t)^{-1}(y_t - x_t)\left((y_t - x_t)^{\top}\theta_t + \epsilon_t(y_t)\right)\right] + \mathbb{E}_t\left[(\beta I + \widehat{\Sigma}_t)^{-1}(y_t - x_t)x_t^{\top}\theta_t\right]$$

$$= \theta_t - \beta(\beta I + \widehat{\Sigma}_t)^{-1}\theta_t + \mathbb{E}_t\left[(\beta I + \widehat{\Sigma}_t)^{-1}(y_t - x_t)\epsilon_t(y_t)\right] + (\beta I + \widehat{\Sigma}_t)^{-1}(\widehat{x}_t - x_t)x_t^{\top}\theta_t.$$

Using this we get

$$\left|\left\langle x_t - u, \theta_t - \mathbb{E}_t[\widehat{\theta}_t]\right\rangle\right|$$

$$\leq \beta\left|(x_t - u)^{\top}(\beta I + \widehat{\Sigma}_t)^{-1}\theta_t\right| + \underbrace{\left|(x_t - u)^{\top}\mathbb{E}_t\left[(\beta I + \widehat{\Sigma}_t)^{-1}(y_t - x_t)\epsilon_t(y_t)\right]\right|}_{(\star)}$$

$$+ \left|(x_t - u)^{\top}(\beta I + \widehat{\Sigma}_t)^{-1}(\widehat{x}_t - x_t)x_t^{\top}\theta_t\right|. \qquad (9)$$

We handle $(\star)$ as follows:

$$(\star) = \mathbb{E}_t \left[ \sqrt{(x_t - u)^\top (\beta I + \widehat{\Sigma}_t)^{-1}(y_t - x_t)(y_t - x_t)^\top (\beta I + \widehat{\Sigma}_t)^{-1}(x_t - u)\epsilon_t(y_t)^2} \right]$$

$$\leq \sqrt{(x_t - u)^\top (\beta I + \widehat{\Sigma}_t)^{-1} \mathbb{E}_t \left[ \epsilon_t(y_t)^2 (y_t - x_t)(y_t - x_t)^\top \right] (\beta I + \widehat{\Sigma}_t)^{-1}(x_t - u)}$$

$$\leq \epsilon \sqrt{(x_t - u)^\top (\beta I + \widehat{\Sigma}_t)^{-1} \widehat{\Sigma}_t (\beta I + \widehat{\Sigma}_t)^{-1}(x_t - u)}$$

$$\leq \epsilon \|x_t - u\|_{(\beta I + \widehat{\Sigma}_t)^{-1}}.$$

Continuing from (9), we get

$$\left| \left\langle x_t - u, \theta_t - \mathbb{E}_t[\widehat{\theta}_t] \right\rangle \right|$$

$$\leq \beta \|x_t - u\|_{(\beta I + \widehat{\Sigma}_t)^{-1}} \|\theta_t\|_{(\beta I + \widehat{\Sigma}_t)^{-1}} + \epsilon \|x_t - u\|_{(\beta I + \widehat{\Sigma}_t)^{-1}} + \|x_t - u\|_{(\beta I + \widehat{\Sigma}_t)^{-1}} \|\widehat{x}_t - x_t\|_{(\beta I + \widehat{\Sigma}_t)^{-1}}$$

$$\leq \sqrt{d}\beta \|x_t - u\|_{(\beta I + \widehat{\Sigma}_t)^{-1}} + \epsilon \|x_t - u\|_{(\beta I + \widehat{\Sigma}_t)^{-1}} + \sqrt{\frac{1}{\beta}} \|x_t - u\|_{(\beta I + \widehat{\Sigma}_t)^{-1}} \|\widehat{x}_t - x_t\|_2,$$

where in the last inequality we use that $\|y_t - x_t\|_{(\beta I + \widehat{\Sigma}_t)^{-1}} \leq 2\|y_t - x_t\|_{\Sigma_t^{-1}}$ by Lemma 9 and the assumption that $\|\theta_t\|_2 \leq \sqrt{d}$.

By Lemma 9 we have $\|x_t - \widehat{x}_t\| = \|\mathbb{E}_{y \sim q_t}[y] - \mathbb{E}_{y \sim \widehat{q}_t}[y]\| \leq \sqrt{d} d^{-5} T^{-10}$. By the choice of $\beta = d^{-2} T^{-4}$, we can bound the expectation of the sum of the last expression as

$$\textbf{Bias} \leq \mathbb{E}\left[ \sum_{t=1}^T \|x_t - u\|_{(\beta I + \widehat{\Sigma}_t)^{-1}} \left( \epsilon + \frac{1}{T^2} \right) \right].$$

**Bounding Bonus term** Notice that with the equivalence established in Lemma 8, our bonus term $b_t$ can also be written as

$$b_t = -8 \left( \epsilon + \frac{1}{T^2} \right) \nabla F(x_t),$$

where $F(\cdot)$ is the entropic barrier on $\mathcal{A}$. Using Lemma 7 and the fact that $F(\cdot)$ is an $O(d)$-self-concordant barrier, we can bound

$$\langle x_t - u, b_t \rangle \leq -\frac{(8\epsilon + 8/T^2)}{4} \|x_t - u\|_{\nabla^2 F(x_t)} + 6\epsilon\nu$$

$$= -\left( 2\epsilon + \frac{2}{T^2} \right) \|x_t - u\|_{\nabla^2 F(x_t)} + O(d\epsilon)$$

$$= -\left( 2\epsilon + \frac{2}{T^2} \right) \|x_t - u\|_{\Sigma_t^{-1}} + O(d\epsilon)$$

$$\leq -\left( \epsilon + \frac{1}{T^2} \right) \|x_t - u\|_{\widehat{\Sigma}_t^{-1}} + O(d\epsilon). \qquad \text{(by Lemma 9)}$$

Thus,

$$\textbf{Bonus} \leq \mathbb{E}\left[ -\sum_{t=1}^T \left( \epsilon + \frac{1}{T^2} \right) \|x_t - u\|_{\widehat{\Sigma}_t^{-1}} + O\left( dT\epsilon \right) \right].$$

**Adding up all terms** Adding up the three terms, we get

$$\mathbb{E}\left[ \sum_{t=1}^T \langle y_t - u, \theta_t \rangle \right] \leq \frac{d \log T}{\eta} + O\left( \eta d\gamma^2 \mathbb{E}\left[ \sum_{t=1}^T c_t^2 \right] + dT\epsilon \right).$$

By the assumption $c_t \in [0, 1]$ and the assumption $\left| \mathbb{E}_t[c_t] - y_t^\top \theta_t \right| \leq \epsilon$, we can further bound the right-hand side by

$$O\left( \frac{d \log T}{\eta} + \eta d\gamma^2 \mathbb{E}\left[ \sum_{t=1}^T c_t \right] + dT\epsilon \right) \leq O\left( \frac{d \log T}{\eta} + \eta d\gamma^2 \mathbb{E}\left[ \sum_{t=1}^T \langle y_t, \theta_t \rangle \right] + (d + \eta d\gamma^2)T\epsilon \right).$$

Then, by rearranging, we find that

$$\mathbb{E}\left[\sum_{t=1}^{T} \langle y_t - u, \theta_t \rangle\right] \leq O\left(\frac{d \log T}{\eta} + \eta d\gamma^2 \sum_{t=1}^{T} \langle u, \theta_t \rangle + dT\epsilon\right).$$

Choosing the optimal $\eta$, we further bound

$$\mathbb{E}\left[\sum_{t=1}^{T} \langle y_t - u, \theta_t \rangle\right] = O\left(d\gamma\sqrt{(\log T)\sum_{t=1}^{T} \langle u, \theta_t \rangle} + dT\epsilon\right) = \tilde{O}\left(d\sqrt{T} + dT\epsilon\right). \qquad (10)$$

By Definition 1, this is a $\sqrt{d}$-misspecification-robust algorithm.

$\square$

## D.2 Regret bound of LCB

*Proof of Theorem 3.* Let $\widehat{\Pi}$ be the set of linear policies created from the vertices of $\widehat{\Omega}$, and let $\Pi$ be set of all linear policies.

The regret bound guaranteed by the $\alpha$-misspecification-robust linear bandit problem is

$$\mathbb{E}\left[\sum_{t=1}^{T} \langle y_t, \theta_t \rangle\right] \leq \min_{y \in \widehat{\Omega}} \sum_{t=1}^{T} y^\top \theta_t + \tilde{O}\left(d\sqrt{T} + \alpha\sqrt{d}\epsilon T\right) \qquad (11)$$

for some $\alpha \geq 1$. By Lemma 2, we have

$$y_t = \mathbb{E}_{\mathcal{A} \sim \widetilde{\mathcal{D}}}[\pi_t(\mathcal{A})].$$

We further define

$$z_t = \mathbb{E}_{\mathcal{A} \sim \mathcal{D}}[\pi_t(\mathcal{A})], \qquad z^\star = \mathbb{E}_{\mathcal{A} \sim \mathcal{D}}[\pi^\star(\mathcal{A})], \qquad y^\star = \mathbb{E}_{\mathcal{A} \sim \widetilde{\mathcal{D}}}[\pi^\star(\mathcal{A})],$$

where $\pi^\star \in \Pi$ is the final regret comparator. Define

$$\epsilon = 4\sqrt{\frac{d \log(NKT/\delta)}{N}},$$

where $N$ is the number of contexts used to construct $\widehat{\Omega}$. Then by Lemma 1 we have $|\langle y_t, \theta_t \rangle - \langle z_t, \theta_t \rangle| \leq \epsilon$ and $|\langle y^\star, \theta_t \rangle - \langle z^\star, \theta_t \rangle| \leq \epsilon$ with probability at least $1 - \delta$ for all $t$. Choosing $\delta = \frac{1}{T^2}$, we obtain

$$\begin{aligned}
\mathbb{E}\left[\sum_{t=1}^{T} \langle z_t, \theta_t \rangle\right] &\leq \mathbb{E}\left[\sum_{t=1}^{T} \langle y_t, \theta_t \rangle\right] + T\epsilon \\
&\leq \sum_{t=1}^{T} \langle y^\star, \theta_t \rangle + \tilde{O}\left(d\sqrt{T} + \alpha\sqrt{d}T\epsilon\right) \qquad \text{(by (11))} \\
&\leq \sum_{t=1}^{T} \langle z^\star, \theta_t \rangle + \tilde{O}\left(d\sqrt{T} + \alpha\sqrt{d}T\epsilon\right) \\
&= \sum_{t=1}^{T} \langle z^\star, \theta_t \rangle + \tilde{O}\left(d\sqrt{T} + \alpha dT\sqrt{\frac{\log(NKT)}{N}}\right).
\end{aligned}$$

This proves the theorem.

$\square$

# E Omitted Details in Section 5

*Proof of Theorem 4.* This is by the same proof as Theorem 2, just noticing that it actually achieves a small-loss bound in (10).

$\square$

*Proof of Theorem 5.* This is similar to the proof of Theorem 3, except that we replace (11) by

$$\mathbb{E}\left[\sum_{t=1}^{T}\langle y_t, \theta_t\rangle\right] \leq \min_{y\in\widehat{\Omega}}\sum_{t=1}^{T}y^{\top}\theta_t + \tilde{O}\left(d\sqrt{\sum_{t=1}^{T}\langle y, \theta_t\rangle} + \alpha\sqrt{d\epsilon}T\right). \tag{12}$$

Following the same steps, we get

$$\mathbb{E}\left[\sum_{t=1}^{T}\langle z_t, \theta_t\rangle\right] \leq \sum_{t=1}^{T}\langle z^{\star}, \theta_t\rangle + \tilde{O}\left(d\sqrt{\sum_{t=1}^{T}\langle z^{\star}, \theta_t\rangle} + \alpha dT\sqrt{\frac{\log(NKT)}{N}}\right), \tag{13}$$

where $z_t = \mathbb{E}_{\mathcal{A}\sim\mathcal{D}}\left[\pi_t(\mathcal{A})\right]$ and $z^{\star} = \mathbb{E}_{\mathcal{A}\sim\mathcal{D}}\left[\pi^{\star}(\mathcal{A})\right]$.

Since the learner is given simulator access, she can draw $N = \tilde{\Omega}(\alpha^2 d^2 T^2)$ samples and make the last term in (13) be a constant. This will give a final regret bound of $\tilde{O}\left(d\sqrt{\sum_{t=1}^{T}\langle z^{\star}, \theta_t\rangle}\right) = \tilde{O}(d\sqrt{L^{\star}})$. $\qquad\square$

