# OpenReview forum: "An Improved Algorithm for Adversarial Linear Contextual Bandits via Reduction"
_NeurIPS.cc/2025/Conference — NeurIPS 2025 poster_

### Official Review · Reviewer_VCk3 · 2025-06-10

**Clarity:** 3
**Significance:** 2
**Originality:** 3
**Rating:** 4
**Confidence:** 2

**Summary:**

This paper studies the linear contextual bandit problem, where contexts (action sets) are generated stochastically and linear losses are selected adversarially. The main contribution is an efficient algorithm that achieves a regret bound of $\tilde O(d^2\sqrt{T})$ for the dimensionality $d$ of action sets and time horizon $T$. The algorithm runs in polynomial time in $d$, $T$, and the number of constraints that specify the action sets, thereby allowing for efficient learning in large action spaces. Moreover, given access to the context simulator, the bound is improved to $\tilde O(d\sqrt{L^*})$, where $L^*$ is the cumulative loss of the best policy. The main technical idea is to reduce the problem to the adversarial linear bandit problem with misspecification. Combing this with a uniform convergence argument and the complexity analysis techniques known in linear programming leads to the desired result.

**Questions:**

1. In equation (1), how to detail with possibly multiple minimizers?
2. In the equation after line 121, what is the rightmost expectation taken about?
3. I could not understand the rightmost expression of $\hat \Omega$ after line 125. Why does it contain neither $\pi$ nor $\Pi$?
4. In line 194, what is the specific amount of the misspecification $\varepsilon$? Does it come from the uniform convergence bound?

Minor comment:
- In line 121, $y_t$ should be $a_t$ for the consistency with the subsequent equation.

**Ethical Concerns:**

["NO or VERY MINOR ethics concerns only"]

**Final Justification:**

The paper presents the first $\mathrm{poly}(d)\sqrt{T}$-regret algorithm for the adversarial linear contextual bandit problem, based on a novel reduction to corruption-robust adversarial linear bandits. I believe this contribution is significant enough to be accepted, although the efficiency in practice appears to have room for improvement.

**Limitations:**

yes

**Paper Formatting Concerns:**

No formatting concerns.

**Quality:**

2

**Strengths And Weaknesses:**

### Strengths
1. This paper provides the first efficient $\mathrm{poly}(d)\sqrt{T}$-regret algorithm for the adversarial linear contextual bandit problem that can handle large action spaces, resolving the open question raised by Liu et al. (2023).
2. The main reduction idea appears novel and interesting, which may inspire future research in this area.

### Weaknesses
1. The details of the misspecification-robust adversarial linear bandit algorithm are deferred to the appendix, and little intuition regarding the proof of Theorem 2 is provided in the main text. This makes it difficult to assess the significance of the adversarial linear bandit algorithm, particularly for readers unfamiliar with the misspecification-robust setting, such as myself.
2. I do not consider this a major weakness, but the algorithm relies on somewhat costly operations such as the ellipsoid method. Additionally, the complexity also depends on the number of constraints, which may be large in some cases. The authors are aware of these points and have discussed them in the paper.

---

> ### Author Rebuttal · Authors · 2025-07-30
>
> We thank the reviewer for their time and for providing an insightful review. Please see our responses below.
>
> **Q1**:   The details of the misspecification-robust adversarial linear bandit algorithm are deferred to the appendix, and little intuition regarding the proof of Theorem 2 is provided in the main text. This makes it difficult to assess the significance of the adversarial linear bandit algorithm, particularly for readers unfamiliar with the misspecification-robust setting.
>
> **A1**:  Thanks for the suggestion. We will add more explanation for Algorithm 2 in the revision. While its guarantee is new, its main algorithmic elements are from prior work.  It is a simplified and improved version of Liu et al. (2024)’s Algorithm 3 (the simplification and the improvement in the bound are possible because in our case the amount of misspecification $\epsilon$ is the same in all rounds). Liu et al. (2024)’s algorithm, in turn, is a combination of the clipped continuous exponential weights of Ito et al. (2020) that guarantees optimal small-loss bound with polynomial computational complexity, and the bonus design in Zimmert and Lattimore (2022) that ensures misspecification robustness (though their original goal is high-probability bound).
>
> Liu, Tajdini, Wagenmaker, Wei.  Corruption-Robust Linear Bandits: Minimax Optimality and Gap-Dependent Misspecification. 2024.
>
> Ito, Hirahara, Soma, Yoshida.  Tight First- and Second-Order Regret Bounds for Adversarial Linear Bandits.  2020.
>
> Zimmert and Lattimore.  Return of the bias: Almost minimax optimal high probability bounds for adversarial linear bandits.  2022.
>
> &nbsp;
>
> **Q2**:   In equation (1), how to detail with possibly multiple minimizers?
>
> **A2**:   Our results allow ties to be broken arbitrarily. The regret analysis is agnostic to the tie-breaking rule used. We will clarify this in the revision.
>
> &nbsp;
>
> **Q3**:   In the equation after line 121, what is the rightmost expectation taken about?
>
> **A3**:   The rightmost expectation is taken over the randomness in $\pi_t$ (i.e., the random policy selected by the algorithm at time $t$).
>
> &nbsp;
>
> **Q4**:  I could not understand the rightmost expression of $\hat{\Omega}$ after line 125. Why does it contain neither $\pi$ nor $\Pi$?
>
> **A4**:  This can be seen by the following equalities.  For simplicity, let’s first assume $\mathcal{A}_1, \ldots, \mathcal{A}_N$, which are drawn i.i.d. from $\mathcal{D}$, are all distinct.
>
> $\hat{\Omega}$
>
> $= \lbrace \hat{\Psi}(\pi) \mid   \pi\in\Pi \rbrace$
>
> $= \lbrace \mathbb{E}_{\mathcal{A}\sim \tilde{D}} [\pi(\mathcal{A})]  \mid  \pi\in\Pi  \rbrace$  &nbsp;&nbsp; &nbsp;&nbsp;  // by the definition of $\hat{\Psi}$, where $\tilde{D}$ is the empirical context distribution
>
> $= \lbrace \frac{1}{N} \sum_{i=1}^N   \pi(\mathcal{A}_i)  \mid  \pi\in\Pi  \rbrace$
>
> $=  \lbrace \frac{1}{N} \sum_{i=1}^N   a_i   \mid  a_i \in \text{conv}(\mathcal{A}_i)  \rbrace$
>
> The key is in the last inequality:  Recall that $\pi$ is a mapping from an action set $\mathcal{A}$ to an element in conv($\mathcal{A}$), and $\Pi$ contains all such mappings. When $\mathcal{A}_1, \ldots, \mathcal{A}_N$ are all different, we can freely choose $\pi(\mathcal{A}_i)$ as an element $a_i \in \text{conv}(\mathcal{A}_i)$ for every $i$. This is essentially the last expression.
>
> If there exist $i\neq j$ such that $\mathcal{A}_i=\mathcal{A}_j$, then we have the additional constraint that $a_i= \pi(\mathcal{A}_i) = \pi(\mathcal{A}_j)=a_j$.
>
> &nbsp;
>
> **Q5**:  In line 194, what is the specific amount of the misspecification $\epsilon$? Does it come from the uniform convergence bound?
>
> **A5**:   Yes, $\epsilon$ comes from the uniform convergence bound.  Its value is essentially the right hand side of Line 132 with $b=1$ and $N=\Theta(t)$ (the number of contexts the learner has collected up to time $t$).
>
> &nbsp;
>
> Thank you for pointing out the typo in the minor comment.

---

> > ### Comment · Reviewer_VCk3 · 2025-08-02
> >
> > Thank you for the detailed responses. While I feel that minor weaknesses in presentation and practicality remain, the authors have adequately addressed my questions. I will keep my original score for now, leaning toward acceptance.

---

### Official Review · Reviewer_Jz1R · 2025-06-26

**Clarity:** 4
**Significance:** 3
**Originality:** 3
**Rating:** 5
**Confidence:** 3

**Summary:**

This paper focuses on the problem of adversarial linear bandits with stochastic action set (as context). Specifically, without the knowledge of the context distribution, the authors provide an algorithm achieving $O(d^2\sqrt{T})$ regret with runtime polynomial in the dimension $d$ and the horzion $T$ (independent of the number of action $K$ in each action set). Moreover, with the knowledge of the context distribution, the authors further show that an improved first-order bound $O(d\sqrt{L_0})$ is achievable, where $L_0$ is the expected loss of the best policy in hindsight. The algorithm is motivated by the recent advance in reducing contextual linear bandits to non-contextual bandits by Hanna et al., 2023. However, the reduction introduced in this paper is slightly different from the one in Hanna et al., 2023 in the sense that Hanna et al., 2023 only considers the stochastic loss case.

**Questions:**

- Based on my understanding, the reason why first-order regret bound is not obtained for the unknown context distribution setting is that the action set approximation error can not be related to the loss of the algorithm. I wonder whether there are some methods to address this?

**Ethical Concerns:**

["NO or VERY MINOR ethics concerns only"]

**Final Justification:**

I read the rebuttal as well as the other reviewers' comment. I decide to maintain my positive score.

**Limitations:**

Yes.

**Paper Formatting Concerns:**

None.

**Quality:**

3

**Strengths And Weaknesses:**

Strength:
- The problem is well-motivated and this paper provides the first algorithm that achieves $O(\sqrt{T})$ regret with runtime independent of the number of action without the knowledge of the context distribution. The paper is also well-written in general.
- The proposed algorithm is intuitive and easy to follow. Specifically, this paper proposes a variant of contextual-to-non-contexutal reduction from Hanna et al., 2023 to handle the adversarial linear loss, which is interesting to me. In addition, to achieve the efficiency improvements, the authors also proposed an algorithm for non-contextual linear bandits with known misspecification, which utilizes the nice property of this reduction.
- With the knowledge of the context distribution, the algorithm does not need to approximate the non-contextual action set and achieves a better $d\sqrt{L_0}$ regret.

Weaknesses:
- As the authors mentioned, the $d$ dependency in the regret bound when the context distribution is unknown is still suboptimal.
- Maybe the authors can add more explanations to Algorithm 2 in the main text. Currently, it seems that the authors just list Algorithm 2 and its guarantee without introducing more about the intuition and the procedure.

In general, I do not find more major concerns.

---

> ### Author Rebuttal · Authors · 2025-07-30
>
> We thank the reviewer for their time and for providing an insightful review. Please see our responses below.
>
> **Q1**:   Maybe the authors can add more explanations to Algorithm 2 in the main text. Currently, it seems that the authors just list Algorithm 2 and its guarantee without introducing more about the intuition and the procedure.
>
> **A1**:  Thanks for the suggestion. We will add more explanation for Algorithm 2 in the revision. While its guarantee is new, its main algorithmic elements are from prior work.  It is a simplified and improved version of Liu et al. (2024)’s Algorithm 3 (the simplification and the improvement in the bound are possible because in our case the amount of misspecification $\epsilon$ is the same in all rounds). Liu et al. (2024)’s algorithm, in turn, is a combination of the clipped continuous exponential weights of Ito et al. (2020) that guarantees optimal small-loss bound with polynomial computational complexity, and the bonus design in Zimmert and Lattimore (2022) that ensures misspecification robustness (though their original goal is high-probability bound).
>
> Liu, Tajdini, Wagenmaker, Wei.  Corruption-Robust Linear Bandits: Minimax Optimality and Gap-Dependent Misspecification. 2024.
>
> Ito, Hirahara, Soma, Yoshida.  Tight First- and Second-Order Regret Bounds for Adversarial Linear Bandits.  2020.
>
> Zimmert and Lattimore.  Return of the bias: Almost minimax optimal high probability bounds for adversarial linear bandits.  2022.
>
> &nbsp;
>
> **Q2**:   Based on my understanding, the reason why first-order regret bound is not obtained for the unknown context distribution setting is that the action set approximation error can not be related to the loss of the algorithm. I wonder whether there are some methods to address this?
>
> **A2**: To make the action set approximation error related to the loss of the algorithm, we essentially need a counterpart of Bernstein inequality for our Lemma 1 (the concentration bound based on Rademacher complexity). There are some related tools in the literature, such as local Rademacher complexity (Bartlett et al., 2005) and offline Rademacher complexity (Liang et al., 2015).  However, we haven’t found a way to integrate them into our framework and left it as a future work.  We will provide more discussions on this in the revision.
>
> Bartlett, Bousquet, Mendelson.  Local Rademacher complexities. 2005.
>
> Liang, Rakhlin, Sridharan. Learning with Square Loss: Localization through Offset Rademacher Complexity.  2015.

---

> > ### Comment · Reviewer_Jz1R · 2025-08-05
> >
> > I thank the authors for their detailed rebuttal. My questions are addressed well and I tend to keep my current score.

---

### Official Review · Reviewer_Aqe6 · 2025-06-29

**Clarity:** 2
**Significance:** 3
**Originality:** 3
**Rating:** 5
**Confidence:** 3

**Summary:**

The work investigates the problem of adversarial linear contextual bandits, where in each round a set $\mathcal{A}_t$ of $K$ actions is drawn i.i.d. from an unknown but fixed distribution over $\mathbb{R}^d$. The randomized learner picks $a_t \in \mathcal{A}_t$ and observes a noisy loss whose expectation is $\langle a_t,\theta_t \rangle \in [-1,1]$, where $\theta_t\in\mathbb{R}^d$ is a loss vector chosen by an oblivious adversary. The regret is defined with respect to the optimal policy, which is known to be linear. The authors prove regret bounds of the order of $d^2\sqrt{T}$ using an algorithm whose running time is polynomial in $d$, $T$, and the number of linear constraints sufficient to define the convex hull of each action set. This is the best regret among all known efficient algorithms with full bandit feedback, no access to simulators, and no dependence on $K$. The lack of dependence on $K$ is useful for applications to certain instances of combinatorial bandits such as shortest paths. When a simulator for the distribution is available, the regret can be improved to order of $d\sqrt{L^\star}$ (up to log factors), where where $L^\star$ is the cumulative loss of the best policy.

**Questions:**

1. The optimal regret $d\sqrt{T}$ is currently achieved either inefficiently or with a polynomial number of calls to a simulator. The question whether the optimal regret could be achieved in poly time (even allowing a dependence on $K$) and without using a simulator seems rather fundamental and is not directly addressed by your work. Could your techniques provide any insight on this question?

2. Is your algorithm the first one working in the convex hull of the action set? What did prevent previous approaches to depend on $C$ rather than $K$?

3. There is a large body of literature proposing algorithms that simultaneosly achieve optimal (or near-optimal) stochastic and adversarial regret bounds for bandits (best of both worlds). Could your techniques be applied to obtain results in this direction?

**Ethical Concerns:**

["NO or VERY MINOR ethics concerns only"]

**Final Justification:**

I had an initial positive opinion on this submission. The opinion did not change after the rebuttal, which clarified some of the aspects in the manuscript. If accepted, I would encourage the authors to use the extra space to add the clarifications provided in the rebuttal.

**Limitations:**

Yes

**Quality:**

3

**Strengths And Weaknesses:**

Strengths: The result is obtained by a novel and technically involved extension to an adversarial setting of a reduction previously proposed for the stochastic setting. As the reduction is to adversarial linear bandits with fixed action set and misspecification, the authors also design a new algorithm for this setting with the required tolerance and runnning time. The lack of the dependence on $K$ allows to obtain the first efficient algorithm with $\sqrt{T}$ regret for combinatorial bandits with random action sets.

Weaknesses: The paper is not particularly easy to understand unless one has a good prior knowledge of the setting and its basic properties (the quality of the writing could also be improved). The abstract suggests that the proposed algorithm allows to solve any combinatorial bandit problem in polytime, whereas this seems true only for those action sets whose convex hull can be described by a polynomial number of linear constraints. Despite the rather impressive technical effort, the overall significance of the contribution remains a bit unclear. The algorithm is neither simple nor practical.

A rather typical NeurIPS contribution that privileges technical prowess over conceptual novelty. It is nevertheless a solid contribution that advances the state of the art. I am therefore supporting acceptance.

---

> ### Author Rebuttal · Authors · 2025-07-30
>
> We thank the reviewer for their time and for providing an insightful review. Please see our responses below.
>
> **Q1**:  The optimal regret $d\sqrt{T}$  is currently achieved either inefficiently or with a polynomial number of calls to a simulator. The question whether the optimal regret could be achieved in poly time (even allowing a dependence on $K$) and without using a simulator seems rather fundamental and is not directly addressed by your work. Could your techniques provide any insight on this question?
>
> **A1**:  This is indeed a fundamental question. Unfortunately, our techniques don't directly resolve the gap between the regret with and without simulators.
>
> The suboptimality in both our bound and Liu et al. (2023) stems from the same core issue: we approximate the true context distribution using empirical samples collected before time $t$, then evaluate the learner's policy on this empirical distribution. Since the empirical samples and the learner's policy are correlated, we must use a uniform convergence bound over all policies, which introduces the extra $d$ factor.
>
> For the special case where action sets are subsets of the standard basis in $\mathbb{R}^K$ (i.e., sleeping bandits), Schneider and Zimmert (2023) developed a sophisticated algorithm to avoid this correlation and achieve the optimal $\sqrt{KT}$ bound. It is not yet clear how to generalize their approach to our setting.  We will add this reference and discuss it in our revision.
>
> Schneider and Zimmert. Optimal cross-learning for contextual bandits with unknown context distributions. 2023.
>
> &nbsp;
>
> **Q2**:  Is your algorithm the first one working in the convex hull of the action set? What did prevent previous approaches to depend on $C$ rather than $K$?
>
> **A2**:  Yes, our algorithm is the first to achieve computational complexity depending only on $C$ rather than $K$.
>
> All previous works studying this problem are built on the approach of (Neu and Olkhovskaya, 2020), which runs an individual linear bandit algorithm on each possible action set, with the loss estimators shared between action sets.  Except for Liu et al. (2023), all works use exponential weights over actions, whose computational complexity scales with $K$.
> Liu et al. (2023) uses FTRL with a log-determinant barrier (logdet-FTRL), whose computational complexity still scales polynomially with $K$.
>
> We initially attempted to modify Liu et al. (2023)'s approach by substituting logdet-FTRL with a linear bandit algorithm whose computation does not scale with K (e.g., SCRiBLe and continuous exponential weights), but that failed: for technical reasons, these algorithms do not seem to allow using shared loss estimators among action sets.
>
> In this paper, we adopt a different approach that runs linear bandit over a "single action set" constructed by averaging up the empirical contexts. This allows us to run SCRiBLe or continuous exponential weights on it without incurring the previous issue.
>
> &nbsp;
>
> **Q3**:  There is a large body of literature proposing algorithms that simultaneously achieve optimal (or near-optimal) stochastic and adversarial regret bounds for bandits (best of both worlds). Could your techniques be applied to obtain results in this direction?
>
> **A3**:  For adversarial contextual bandits with stochastic contexts, there is indeed work investigating best-of-both-worlds guarantees under the assumption of known context distribution or access to a simulator (Kato and Ito, 2023; Kuroki et al., 2024). In this case, our approach can also achieve best-of-both-worlds results by simply reducing our problem to a linear bandit algorithm with best-of-both-worlds guarantees.
>
> However, when the context distribution is unknown and there is no simulator, our current technique faces a limitation: the misspecification parameter $\epsilon=\tilde{\Theta}(1/\sqrt{t})$ introduced in our reduction to linear bandits causes the regret to scale as $\sqrt{T}$ even in stochastic environments, preventing us from achieving the $\log(T)$ regret characteristic of the stochastic setting. Resolving this would be an interesting future direction.
>
> Kato, Ito. Best-of-Both-Worlds Linear Contextual Bandits. 2023.
>
> Kuroki, Rumi, Tsuchiya, Vitale, Cesa-Bianchi.  Best-of-Both-Worlds Algorithms for Linear Contextual Bandits.  2024.

---

> > ### Comment · Reviewer_Aqe6 · 2025-08-01
> >
> > Thanks for your answers to my questions. I am standing by my original score while I wait for comments from other reviewers.

---

### Official Review · Reviewer_aqgX · 2025-07-01

**Clarity:** 2
**Significance:** 2
**Originality:** 2
**Rating:** 4
**Confidence:** 3

**Summary:**

The paper studies the adversarial linear contextual bandits problem with stochastic action sets.
In this problem, in each round, an adversary chooses a loss vector $\theta_t \in R^d$
and a random action set (here also referred to as "context")
$A_t \subseteq R^d$ is chosen via sampling from some fixed (but unknown) distribution.
The algorithm can then choose some vector $a_t \in A_t$, incurring
the loss $\ell_t(a_t) = <a_t, \theta_t>$.
The goal is to compete with the set of mappings from
context (aka action sets) to actions;

The paper obtains an algorithm with
$\tilde{O}(d^2\sqrt{T})$ regret
that runs in time $poly(d, T, C)$ time where
$C$ is the number of linear constraints defining $A_t$.
Importantly, the running time does not depend on the number of actions,
which was a major limitation of the prior work.
As a special case, this gives the first polynomial-time
algorithm with sublinear regret for combinatorial bandits with adversarial losses and stochastic action sets.
This can further be improved to $\tilde{O}(d\sqrt{L^*})$ when
given access to a simulator for sampling $A_t$.

The paper obtains the result by providing a reduction to
the linear bandits with misspecification (where the action set is fixed).
The problem is misspecified in that the feedback received
by the learner is not unbiased; they receive a feedback
that is at most $\epsilon$ away from the inner product of $\theta_t$
and the chosen action.

**Questions:**

Questions:
- In the definition of linear bandits, does the learner
  incur the loss they see as feedback or do they incur $<y_t, \theta_t>$?
- The definition of regret in line 84 seems wrong; there should be an expectation over $A_t$?
  Prior work (e.g., see Liu et al.) also take expectation.
  Without expectations, assuming $A_t$ comes from a distribution that is not discrete,
  there are likely going to be different $A_t$ values across all $t$.
- Line 90: "learner decides": I think you mean adversary decides?
- The reduction seems fairly generic. Are their similar problems in which the authors believe
  such approaches would work? Alternatively, are there similar examples known in the literature the authors are aware of?
  If so it would be good to discuss them in more detail.
- It would be good if you provide a more detailed comparison of your reduction with that of Hanna et al.

**Ethical Concerns:**

["NO or VERY MINOR ethics concerns only"]

**Final Justification:**

The authors have answered my questions and I still lean towards acceptance.

**Limitations:**

Yes

**Quality:**

2

**Strengths And Weaknesses:**

Strengths:
- The paper addresses a computational limitation of existing work.
  As such, it fits in nicely with a larger (and growing) body of literature which have attempted to do this
  for many statistical problems.
- The approach is based on a reduction to a simpler problem, which in my opinion is nice.

Weaknesses:
- The regret does not match that of the existing works which are not computationally efficient, though
  in my opinion this is fine.
- The paper has quiet a few typos (e.g., see below) and could benefit from proofreading.

---

> ### Author Rebuttal · Authors · 2025-07-30
>
> We thank the reviewer for their time and for providing an insightful review. Please see our responses below.
>
> **Q1**:  In the definition of linear bandits, does the learner incur the loss they see as feedback or do they incur $\langle y_t, \theta_t\rangle$?
>
> **A1**:   Our Eq.(2) assumes the learner incurs the loss $\langle y_t, \theta_t\rangle$. However, if instead the learner incurs the observed loss, the regret bounds differ by at most $2\epsilon T$ under our misspecification assumption—a difference that can be absorbed by the last term in Eq.(2). Thus, both interpretations work.
>
> &nbsp;
>
> **Q2**:  The definition of regret in line 84 seems wrong; there should be an expectation over $\mathcal{A}_t$?
>
> **A2**:  Thank you for pointing this out. You are right—the regret definition in line 84 should include an expectation over $\mathcal{A}_t$.  We will correct this in the revision. Note that our theoretical analysis indeed takes expectation over $\mathcal{A}_t$.
>
> &nbsp;
>
> **Q3**:  Line 90: "learner decides": I think you mean adversary decides?
>
> **A3**:  Thank you for catching this.  Yes, it should be "adversary decides".
>
> &nbsp;
>
> **Q4**:  The reduction seems fairly generic. Are there similar problems in which the authors believe such approaches would work? Alternatively, are there similar examples known in the literature the authors are aware of?
>
> **A4**:  Yes, our reduction approach has broader applicability. We will discuss two relevant examples:
>
> First, our problem setting is a special case of MDPs with fixed transitions and adversarial losses, which has been extensively studied recently. Our approach provides a potential solution for such settings when dealing with continuous or exponentially large action spaces.
>
> Second, recent work on linear Markov games (Wang et al. 2023; Dai et al. 2024) reduces the problem to adversarial linear contextual bandits with stochastic action sets. While their regret guarantees are independent of the number of actions $K$, their computational complexity still scales with $K$. Our approach could potentially eliminate this computational dependence on $K$ in their setting as well.
>
> Wang, Liu, Bai, Jin.  Breaking the curse of multiagency: Provably efficient decentralized multi-agent rl with function approximation. 2023.
>
> Dai, Cui, Du.  Refined Sample Complexity for Markov Games with Independent Linear Function Approximation. 2024.
>
> &nbsp;
>
> **Q5**: It would be good if you provide a more detailed comparison of your reduction with that of Hanna et al.
>
> **A5**:  Thanks for the suggestion. We will add an extensive comparison with Hanna et al. in our final version.  While the reductions are conceptually related, the key difference lies in our primary focus on computational efficiency. Their work only focuses on regret bounds, and there are still gaps in making their algorithm polynomial time.
>
> Specifically,  we reduce the problem to linear bandits with domain $\hat{\Omega}$ — the image under $\Psi$ of all policies $\pi$ — which is a polytope with poly($d$, $C$, $T$) linear constraints that allows for linear programming tools.  On the other hand, Hanna et al. reduce the problem to linear bandits with domain $\mathcal{X}$ — the image under $\Psi$ of linear policies $\pi_\phi$ where $\phi$ comes from an $\epsilon$-net of $\mathbb{R}^d$ — which has $T^d$ discretized points. It remains unclear how their algorithm efficiently: 1) constructs and maintains these points and 2) runs a linear bandit over them. While they mention Phased Elimination, this algorithm has computational complexity scaling with the number of actions ($T^d$ in their case).
>
> Dealing with the polytope requires novel techniques, including our method of mapping a point in $\hat{\Omega}$ to policies (Section 3.2), and our analysis that bounds the estimation error using Rademacher complexity (Section 3.3) rather than discretization plus Hoeffding’s inequality and a union bound as in Hanna et al.
>
> Finally, we operate on the adversarial setting while they on the stochastic setting, a strict generalization.

---

### Decision · Program_Chairs · 2025-09-17

**Decision:**

Accept (poster)

**Comment:**

This paper provides an efficient algorithm for adversarial linear bandits with a stochastic arm set, which runs in polynomial time and does not depend on the number of arms. This is achieved by a reduction from adversarial linear contextual bandits to misspecification-robust linear bandits. Although the dependence on $d$ is not yet optimal, all reviewers appreciate the technical contributions of the paper and have given positive evaluations. The authors are encouraged to incorporate the reviewers’ constructive feedback into the revised manuscript.